# Spatiotemporal H₂O₂ flashes coordinate actin cytoskeletal remodeling and regulate cell migration and wound healing

Maurice O'Mara, Suisheng Zhang & Ulla G. Knaus ⬤ ✉

Well-organized repair of damaged barrier epithelia is vital for infection control, resolution of inflammation, and enduring physical protection. Cysteine thiol and methionine oxidation are connected to cytoskeletal rearrangements in cell migration and wound healing, but how localized redox signaling is achieved to regulate dynamic processes remains elusive. Here, we identify DUOX2, a mucosal barrier NADPH oxidase, as vesicle-incorporated H₂O₂ source, localizing to sites of cytoskeletal reorganization, and facilitating tunneling nanotube and lamellipodia formation. Using traceable fluorescent DUOX2 and the membrane-bound H₂O₂ sensor HyPer7-MEM enabled insight into DUOX2 vesicle trafficking and H₂O₂ generation at sites of actin polymerization and dynamic remodeling. Stable expression or ablation confirmed DUOX2 generated H₂O₂ as a catalyst for cell-cell connections, random motility and directed migration. We identify a signaling axis from the mechanosensor PIEZO1 to DUOX2 and FER tyrosine kinase activation to initiate retraction wave-mediated efficient wound closure in epithelial cells, a prerequisite for barrier integrity.

The actomyosin cytoskeleton undergoes constant remodeling in response to mechanical and biochemical cues. Actin filaments and microtubules are essential for cell shape, organelle organization, vesicle trafficking, proliferation, and motility[1]. Cell migration requires comprehensive reorganization of the cytoskeleton, combining protrusive forces at the leading edge with contractile forces at the rear. Redox signaling has long been implicated as an important regulator of motility-associated processes including actin polymerization and depolymerization, F-actin branching, cell protrusion extension, and rear retraction[2,3]. H₂O₂ oxidizes both F-actin and G-actin at C-terminal cysteine residues on α-actin (Cys376) and β-actin (Cys374), affecting the actin polymerization assembly and elongation rate[4]. Oxidation of F-actin Met44 and Met47 by flavin-dependent monooxygenase MICAL leads to depolymerization and severing of actin filaments[5]. An increase in intracellular H₂O₂ activates redox-regulated tyrosine kinases such as SRC, ABL or FER, leading to tyrosine phosphorylation of cortactin, a nucleation-promoting factor that binds to activated Arp3 and stabilizes branched actin networks[6,7]. Many other proteins connected to cytoskeletal remodeling are regulated by oxidation, among others the GTPase RhoA, MHC type II, cofilin-1, and protein phosphatases.

The abundance of cytosolic H₂O₂ degrading enzymes such as peroxiredoxins implies the need for rapid, localized generation of H₂O₂ for spatiotemporal oxidation of signaling molecules. NADPH oxidases (NOX) are the ideal candidate enzymes to fulfill this function due to their compartmentalized localization and the intricate regulation of their catalytic activity. While all seven mammalian NOX enzymes (NOX1-5, DUOX1-2) contain homologous structural elements essential for electron transfer from NADPH to oxygen, their enzyme complex composition, posttranslational modifications, and regulatory mechanisms differ[8]. NADPH oxidases have been implicated in cell migration in different cell types and via different mechanisms[9–13]. In zebrafish and *Drosophila*, the immediate response to wounding induced calcium flashes that activated DUOX by binding to its EF-hand motifs, thereby triggering H₂O₂ generation that acted as a chemoattractant for neutrophils and altered signaling responses[14–16]. In mammals two highly homologous DUOX enzymes exist, both mainly

Conway Institute, School of Medicine, University College Dublin, Dublin, Ireland. ✉e-mail: ulla.knaus@ucd.ie

expressed in the thyroid and mucosal barrier epithelia of the gastrointestinal tract, respiratory system, pancreas, and gall bladder.

In plants calcium-triggered cell-to-cell waves of $H_2O_2$ have been connected to the NOX5-related NADPH oxidase RBOHD[17,18], and this process was documented using fumigated H2DCFDA or optical nanosensors[19,20]. The creation of genetically encoded $H_2O_2$ probes such as redox-sensitive fluorescent protein (roGFP-Orp1) or coupling a bacterial $H_2O_2$ sensing protein (OxyR) to circularly permuted GFP (HyPer) has enabled detection of $H_2O_2$ in real-time[21]. Combining these tools with NOX isoform localization at distinct cellular compartments permits unparalleled insights into spatiotemporal onset and termination of NADPH oxidase-generated $H_2O_2$ signals and their downstream effects. Here, we identify DUOX as an $H_2O_2$ source at distinct membrane compartments, dynamically generating bursts of $H_2O_2$ at extending and retracting actin structures. DUOX2-derived $H_2O_2$ contributes to lamellipodia formation, promotes cell migration and is essential for efficient wound closure and restoration of barrier function by transmitting mechanosensory signals to FER tyrosine kinase and cortactin.

## Results

### Characterization of DUOX2 vesicular trafficking

To investigate how DUOX2 contributes to spatiotemporal $H_2O_2$-regulated cytoskeletal dynamics, lung epithelial NCI-H661 cells that lack DUOX1/2 due to epigenetic silencing and do not express other NADPH oxidases[22,23] were lentivirally transduced with DUOXA2 (control), the essential DUOX2 dimerization partner, and then either with multi-epitope tagged wildtype DUOX2 (DUOX2 WT) or catalytically inactive DUOX2 mutant (DUOX2 E843Q)[24] (Fig. 1a). Expression of DUOX2 WT and mutant (180 kDa), detected by epitope tag (HA) and DUOX2 specific antibodies, was comparable (Fig. 1b). As expected, H661 cells expressing solely DUOXA2 or DUOX2 E843Q/DUOXA2 did not generate $H_2O_2$ upon stimulation, while $H_2O_2$ generation by DUOX2 WT/DUOXA2 was comparable to wildtype or single HA tagged DUOX2 used previously (Fig. 1c)[25]. Internal membrane and cell surface localization of the DUOX2 mutant matched DUOX2 WT (Fig. 1d, Supplementary Fig. 1a). For live cell analysis, we generated fluorescently tagged DUOX2 by inserting the bilirubin-inducible fluorescent protein UnaG[26] close to the N-terminus of DUOX2 (Fig. 1e). UnaG was chosen for its β barrel structure (PDB 4I3B) and smaller size (UnaG 15.6 kDa, GFP 28 kDa). The slightly decreased expression of UnaG-DUOX2 (195 kDa, Fig. 1f) in H661-DUOXA2 cells correlated with a minor drop in $H_2O_2$ generation when compared to HA-DUOX2 (Fig. 1g). The UnaG fluorescence signal colocalized with DUOX2 as detected by anti-DUOX staining in immunofluorescence microscopy (Fig. 1h, Supplementary Fig. 1b), confirming the creation of a fully functional fluorescent DUOX2.

The plasma membrane localization of DUOX2 was described[25,27], but the nature of DUOX2 positive vesicles[25] is unresolved. In live cell imaging we observed continuous trafficking of small UnaG-DUOX2 positive vesicles from internal membranes to the plasma membrane with UnaG-DUOX2 remaining on the cell surface for ~2 min, followed by internalization into larger vesicles (Fig. 1i, Supplementary Movie 1). Live cell imaging showed that these UnaG-DUOX2 vesicles were trafficked to areas of the cell where membrane extension was occurring. We confirmed that these fluorescent vesicles were indeed DUOX2 positive (Fig. 1j). Co-staining of vesicles indicated that Early Endosome Antigen 1 (EEA1) strongly colocalized with DUOX2 on large intracellular vesicles, while smaller vesicles were only DUOX2 positive (Fig. 1k). Further analysis revealed that large DUOX2 vesicles were positive for Sorting Nexin 4 (SNX4) (Fig. 1l) and RAB11 (Supplementary Fig. 1c), both involved in the endosomal recycling pathway. DUOX2 did not colocalize with Lysosomal-associated membrane protein 1 (LAMP1) (Supplementary Fig. 1d), suggesting that large DUOX2 containing vesicles are primarily bound for recycling and not degradation. Quantification of vesicle diameter by imaging analysis confirmed these

distinct DUOX2 positive vesicle populations, with small vesicles ($\sim 6\,\mu m$) transporting Golgi-modified DUOX2[28] to the plasma membrane, and larger vesicles ($\sim 1.4\,\mu m$) being used for internalization and recycling (Fig. 1m, Supplementary Fig. 1e). Vesicular recycling is connected to spatially restricted signaling pathways required for actin rearrangements including the formation of cell protrusions and directed migration, suggesting a role for DUOX2-mediated redox signaling at distinct sites of actin remodeling.

### DUOX2 activity induces tunneling nanotube formation

On closer observation of cell protrusions emanating from H661 cells expressing DUOX2 WT, nascent or fully formed tunneling nanotubes (TNTs) were apparent, especially when using serum deprivation as a trigger. TNTs are intercellular membranous channels formed by cytoskeletal extension containing actin and sometimes acetylated tubulin[29]. Immunofluorescence and live cell imaging revealed the presence of DUOX2 in the initial stages of TNT formation and in vesicles trafficked along mature TNTs. In nascent TNTs DUOX2 localized to the tips of the forming TNT, commonly colocalized with cortactin, an established driver of actin polymerization, and within vesicle "bulges" along the TNT (Fig. 2a; see tip and bulge localization). In mature, cell-cell connected TNTs with an acetylated tubulin backbone, DUOX2 was evident in vesicle-like structures along the TNT (Fig. 2b). In accord, UnaG-DUOX2 localized to the tips of extending TNTs together with cortactin (Fig. 2c). Quantification of mature TNTs in H661 cell lines showed a marked increase of connecting TNTs in the presence of wildtype DUOX2 compared to cells expressing catalytically inactive DUOX2 E843Q mutant or cells without DUOX2 expression (Fig. 2d), TNT formation was also significantly decreased by the addition of the NOX/DUOX inhibitor GKT137831. Further analysis indicated DUOX2 colocalization with the TNT markers RAB11 (Fig. 2e), Myo10 (Fig. 2f) and GAP43 (Fig. 2g) at the tips of extending TNTs. UnaG-DUOX2 containing vesicles traveled along the full length of mature TNTs. Over the course of 90 min, we recorded UnaG-DUOX2 vesicles entering the TNT from one cell and traveling along the TNT to a recipient cell (Fig. 2h,i; Supplementary Movie 2).

In neurons and astrocytes prolonged treatment with non-physiological concentrations of $H_2O_2$ (100–200 μM) induced TNTs[30,31]. Our observations suggest that nanomolar $H_2O_2$, spatiotemporally produced by DUOX2, is a physiological trigger for TNT formation. To visualize localized $H_2O_2$ generation we employed the genetically encoded fluorescent indicator HyPer7, a pH stable, ultrasensitive $H_2O_2$ sensor[32], tagged with the last 23 C-terminal amino acids of H-RAS (HyPer-MEM). This motif includes two palmitoylation sites and the CAAX farnesylation site, thereby targeting Golgi emanating vesicles trafficking to the plasma membrane (Fig. 2j)[33]. Characterization of HyPer7-MEM expressing H661 cell lines showed an increased HyPer7 488/405 nm mean intensity signal in DUOX2 WT cells over the control cell line (Supplementary Fig. 2a). This signal was decreased by the addition of BAPTA-AM or diphenyleneiodonium (DPI), both known inhibitors of DUOX2 activity (Supplementary Fig. 2a, b). $H_2O_2$ specificity of HyPer7-MEM was further validated by comparing mean intensity of the HyPer7 488/405 nm ratio with the $H_2O_2$ insensitive HyPer7 mutant SypHer7-MEM (Supplementary Fig. 2c). These controls confirmed the specificity of the HyPer7-MEM signal in H661 DUOX2 WT cells. Analysis of HyPer7-MEM ratiometric movies illustrated that 488/405 nm pixel intensity peaks were reproduced in the 488 nm channel (Supplementary Fig. 2d, Supplementary Movie 3), suggesting that the 488 nm channel could be used for higher resolution microscopy requiring a longer pixel dwell time and frame acquisition. HyPer7-MEM ratio signals were increased at the leading edge of plasma membrane protrusions in H661 DUOX2 WT cells (Supplementary Fig. 2e). In fixed cells the HyPer7-MEM 488 nm signal peaks overlapped with DUOX2 localization at the plasma membrane (Supplementary Fig. 2f). Acidification of the media did not change the HyPer7-MEM

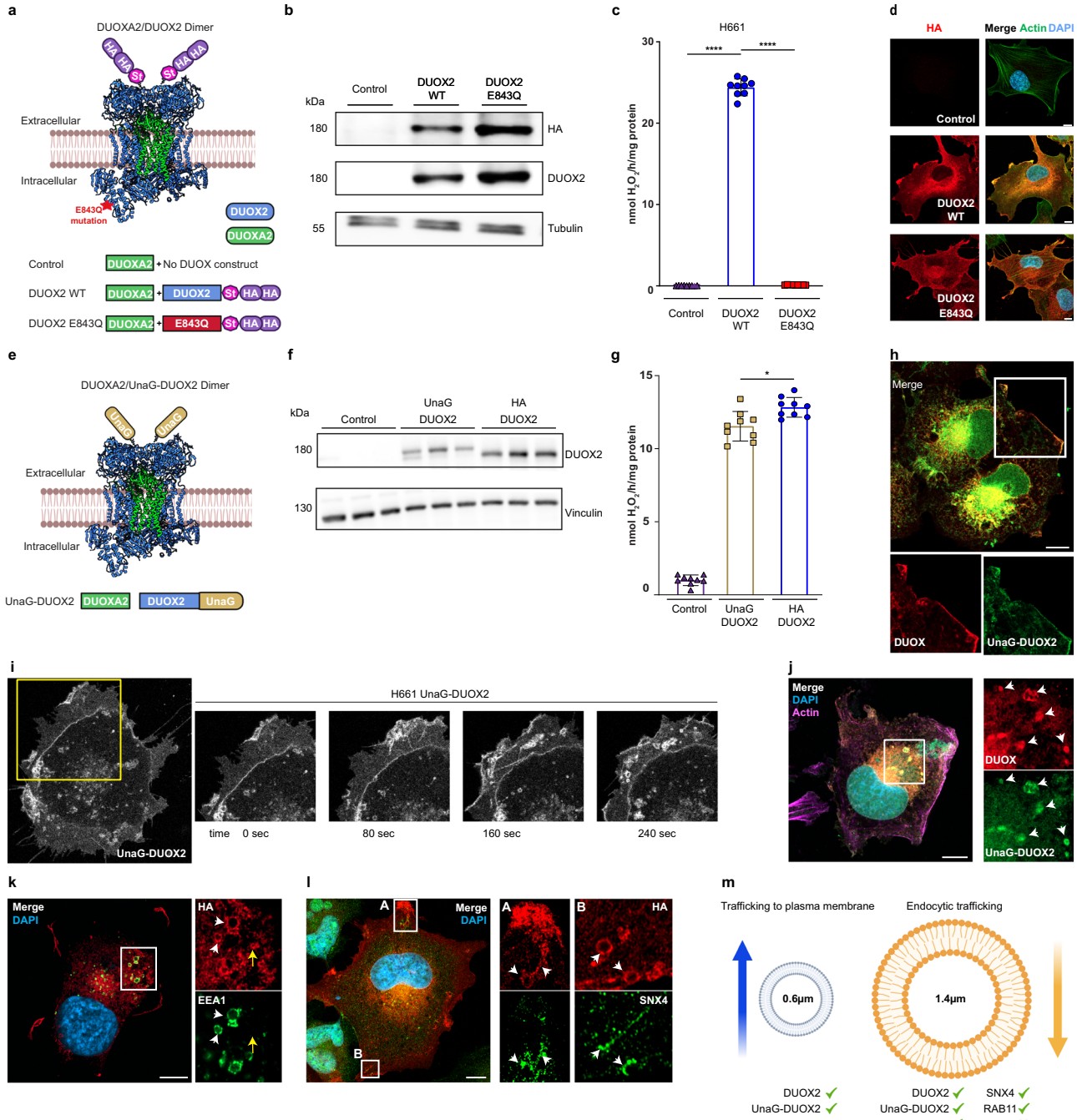

**Fig. 1 | Characterization of DUOX2 vesicular trafficking. a** Schematic of epitope tagged DUOX2 constructs. **b** DUOX2 transgene expression in H661 cell lines confirmed by immunoblot with anti-HA or anti-DUOX2 antibodies; tubulin served as loading control. **c** $H_2O_2$ generation by H661 cells stably expressing DUOXA2 (control) or DUOXA2 in combination with DUOX2 WT or inactive DUOX2 E843Q, stimulated with thapsigargin/PMA (see "Methods"). **d** Localization of DUOX2 WT and mutant by immunofluorescent microscopy (HA, red). **e** Schematic of fluorescently tagged UnaG-DUOX2. **f**, DUOX2 expression in transfected H661 cells using anti-DUOX2 antibody; vinculin served as loading control. **g** $H_2O_2$ generation by H661 cells transiently transfected with DUOXA2 (control), or DUOXA2 in combination with UnaG-DUOX2 or HA-DUOX2, stimulated with thapsigargin/PMA. **h** Colocalization of UnaG-DUOX2 signal (green fluorescence) with anti-DUOX antibody staining (red) in fixed cells; inserts show digital magnification of white box area, white arrows indicate plasma membrane localization of UnaG-DUOX2. **i** Cell surface and vesicular localization of UnaG-DUOX2 in H661 cells; inserts show frames extracted from live cell movie of area denoted by yellow box (Supplementary Movie 1). **j** Co-staining of UnaG-DUOX2 (green) with anti-DUOX antibody

(red) on intracellular vesicles in fixed H661 cells, inserts show digital magnification of area denoted by white box, white arrows indicate vesicles. Immunofluorescence staining of DUOX2 (HA, red) localization in (**k**), early endosomes (EEA1, green) and in (**l**), recycling endosomes (SNX4, green) in H661 cells expressing DUOX2 WT; inserts show digital magnification of area denoted by white box, white arrows indicate vesicles, yellow arrow in (**k**) indicates vesicle with DUOX2 lacking EEA1. **m** Schematic of intracellular DUOX2 vesicle measurements showing colocalized vesicle markers and vesicle sizes (trafficking to plasma membrane 0.6 μm ± 0.3 μm, $n = 89$; endocytic trafficking 1.4 μm ± 0.6 μm, $n = 42$). **c**, **g** Each point represents a single measurement, $n = 3$ independent experiments, data are presented as mean ± SD, Brown-Forsythe and Welch ANOVA with Dunnett's T3 post hoc test; *$P < 0.05$, ****$P < 0.0001$. Co-stained with phalloidin for actin (**d** green, **j** purple) and DAPI (blue). **d**, **h**, **j**, **k**, **l** Scale bars 10 μm, microscopy analyses representative of $n = 5$. **a**, **e**, **m** Created in BioRender. Knaus, U. (2025) https://BioRender.com/xgrzcpe, https://BioRender.com/k6zyosb, https://BioRender.com/mnaww5s, modified in Adobe Illustrator.

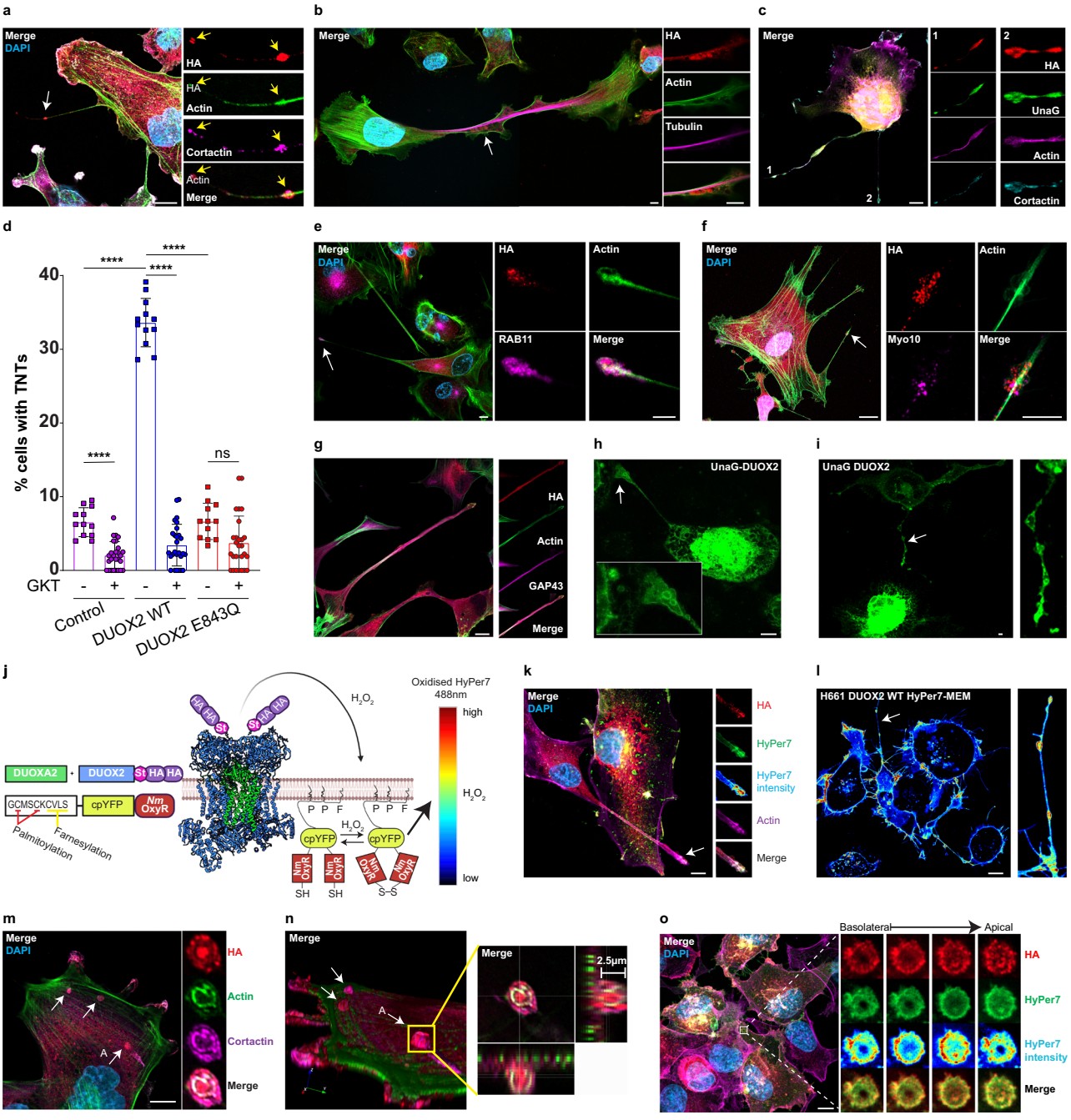

488 nm signal, indicating that the 488 nm signal is pH stable and suitable for intensiometric imaging.

DUOX2 and HyPer7-MEM colocalized at the tips of nascent TNTs (Fig. 2k). The pixel intensity of HyPer7-MEM at the TNT tip (heatmap) indicates substantial H$_2$O$_2$ generation at the point of TNT extension. In live cell imaging H$_2$O$_2$ generation was not only evident at the tip but also in vesicle bulges moving along TNTs (Fig. 2l, Supplementary Movie 4), recapitulating the UnaG-DUOX2 localization in TNTs. In addition to serum deprivation, murine fibroblast conditioned media augmented TNT formation. This treatment also induced protrusions containing DUOX2, cortactin and actin in concentric rings that formed on apical membranes of DUOX2 WT expressing H661 cells (Fig. 2m). 3D rendering of z-stacks indicated an apical extension of ~2.5 μm with DUOX2 generating H$_2$O$_2$ in these ring structures (Fig. 2n, o; colocalization with HyPer7-MEM high pixel intensity peak). DUOX2 containing concentric rings were observed in UnaG-DUOX2 live cell imaging, and

in fixed H661 DUOX2 WT cells co-localizing with the TNT markers RAB11, Myo10 and GAP43 (Supplementary Fig. 2g–j). To distinguish these protrusions from similar actin-based structures, we induced circular dorsal ruffles (CDRs) in DUOX2 WT expressing H661 cells (Supplementary Fig. 2k). DUOX2 was localized to CDRs, but CDRs formed a distinct structure with a hollow interior and larger size than the concentric rings (Supplementary Fig. 2k) and possessed a very short lifetime while apical protrusions persisted. Further, CEP164 and Arl13b, two primary cilia markers, did not localize to DUOX2 positive apical protrusions (Supplementary Fig. 2l, m). To our knowledge, these circular actin ring structures have not yet been reported, but they might be connected to TNT formation.

## Characterization of endogenous DUOX ablation in BxPC3 cells
We wished to compare the findings in H661 cells exogenously expressing active or inactive DUOX2 to a cell type with endogenous

**Fig. 2 | DUOX2 activity induces tunneling nanotube (TNT) formation. a** DUOX2 (HA, red) containing vesicles at tip (insert, yellow arrows) of a nascent TNT in DUOX2 WT expressing H661 cells (actin, green; cortactin, purple). **b** DUOX (red) containing vesicles in mature TNT connecting two DUOX2 WT expressing H661 cells (white arrow), inserts are digital zoom (actin, green; acetylated α-tubulin, purple). Image is stitched from 2 fields of view on 63X oil objective. **c** UnaG-DUOX2 (green) localization to the tips of two nascent TNTs (denoted 1 and 2) in H661 cells, co-stained with anti-DUOX antibody (red), actin (purple) and cortactin (light blue). **d** Quantification of cells with at least one mature connecting TNT in H661 cells expressing DUOXA2 (control), or DUOXA2 combined with DUOX2 WT, or with DUOX2 E843Q, ± 10 μM GKT137831. Each point represents the average of 5 fields of view from 4 experiments, each with 3 replicates, data are presented as mean ± SD, Brown-Forsythe and Welch ANOVA with Dunnett's T3 post hoc test, ****$P < 0.0001$. DUOX2 (DUOX **e**, HA **f**, **g**, red) colocalization with RAB11 (**e**, purple), Myo10 (**f**, purple) and GAP43 (**g**, purple) in nascent TNTs in H661 cells expressing DUOX2 WT. **h** Live cell image of UnaG-DUOX2 vesicles within a TNT in H661 cells, insert shows digital zoom of area denoted by white arrow. **i** Localization of UnaG-DUOX2 within transport vesicles along a TNT during live cell imaging of H661 cells expressing UnaG-DUOX2, insert shows digital zoom of area denoted by white arrow (Supplementary Movie 2). **j** Scheme of membrane embedded HyPer7-MEM

oxidation by DUOX2. **k**, $H_2O_2$ generation at the tip of an extending TNT with HyPer7 signal colocalizing with DUOX2 (HA, red) in fixed DUOX2 WT expressing H661 cells. **l** Live cell confocal video microscopy image of HyPer7 signal peaks in vesicles and at the tip of an unattached TNT in DUOX2 WT expressing H661 cells, inserts show digital zoom of TNT indicated by white arrow (Supplementary Movie 4). **m** DUOX2 (HA, red) localization to apical double ring protrusions with actin (green) and cortactin (purple). White arrows denote apical surface protrusions, digital zoom of arrow A shown in inserts. **n** 3D rendering of z-stack from (**m**), white arrows indicate apical protrusions, yellow box expands to x/y projection of z-stack indicating protrusion distance. **o** Maximum projection z-stack of DUOX2 WT and HyPer7-MEM expressing H661 cells, indicating DUOX2 (HA, red) and HyPer7 (green and heatmap) localization in an apical protrusion. Inserts show a digital zoom of the white box with z-slices visualizing DUOX2 and $H_2O_2$ generation throughout an apical protrusion. **a–c, e–i, k**, Cells serum starved for 24 h before fixation or live cell imaging. **m–o** Cells incubated in murine fibroblast conditioned media for 24 h pre-fixation. **k, o** HyPer7 presented as green for merged image and as a heatmap (HyPer7 pixel intensity) in inserts to visualize $H_2O_2$ peaks. Scale bars **a–c, e–h, k–o** 10 μm, **i** 100 μm, microscopy analyses representative of $n = 5$. **j** Created in BioRender. Knaus, U. (2025) https://BioRender.com/fodhypa, modified in Adobe Illustrator.

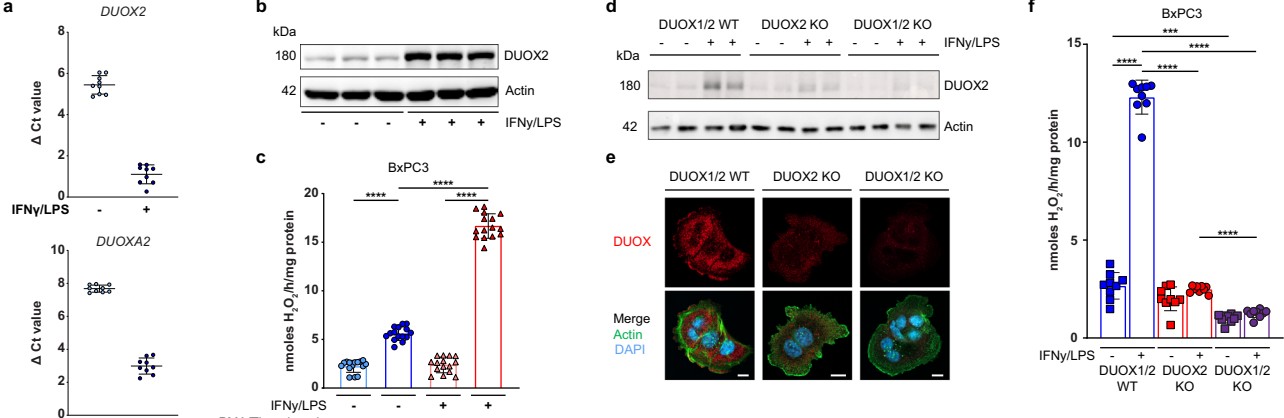

**Fig. 3 | Characterization of endogenous DUOX ablation in BxPC3 cells. a** *DUOX2* and *DUOXA2* expression in untreated or IFNγ (10 ng/ml) / LPS (100 ng/ml) treated (24 h) BxPC3 cells. Each point represents the mean ΔCt ± SD of one measurement from 3 technical replicates, taken from 3 independent experiments. **b** DUOX2 expression in BxPC3 cells with and without IFNγ/LPS pretreatment. **c** $H_2O_2$ generation by BxPC3 cells with and without IFNγ/LPS pretreatment, either unstimulated or after stimulation with PMA/thapsigargin. **d, e** CRISPR-mediated ablation of DUOX2 or DUOX1/2 from DUOX1/2 WT BxPC3 cells shown by **d**, immunoblot in

indicated conditions and **e**, by microscopy (DUOX, red; actin, green; scale bar 10 μm, microscopy analyses representative of $n = 5$). **f** $H_2O_2$ generation by DUOX1/2 WT, DUOX2 KO and DUOX1/2 KO BxPC3 cell lines, ±24 h pre-stimulation by IFNγ/LPS, all conditions stimulated with PMA/thapsigargin. **b, d** Actin served as loading control. **c, f** Data are presented as mean ± SD, Brown-Forsythe and Welch ANOVA with Dunnett's T3 post hoc test; ***$P < 0.0005$, ****$P < 0.0001$, $n = 3$ biological replicates each with 3 technical replicates.

DUOX2 expression. Pancreatic BxPC3 cells express DUOX2 and low levels of DUOX1, but not other NADPH oxidases (our data and[34]). BxPC3 cells display all the hallmarks expected of functional DUOX2 expression including upregulation by IFNγ/LPS (*DUOX2* (Ct 24 → 19), *DUOXA2* (Ct 26 → 21)), which was confirmed by anti-DUOX2 immunoblotting (Fig. 3a, b). Further, low basal $H_2O_2$ generation was highly augmented by PMA/thapsigargin stimulation (Fig. 3c). We next generated both DUOX2 (DUOX2 KO) and combined DUOX1/DUOX2 (DUOX1/2 KO) knockout BxPC3 cell lines using CRISPR. DUOX expression was negligible in CRISPR KO cell lines (immunoblot, Fig. 3d), and while endogenous DUOX (DUOX1/2 WT) localized to the plasma membrane as expected, membrane localization was absent in both DUOX2 KO and DUOX1/2 KO cell lines (Fig. 3e). $H_2O_2$ production was significantly decreased in DUOX2 KO cells (Fig. 3f, red) with a further decrease after additional DUOX1 knockout (Fig. 3f, purple). Serum-starved BxPC3 cells expressing DUOX1/2 formed TNTs at a lower rate than DUOX2 WT expressing H661 cells, yet endogenous

DUOX also localized to both nascent and mature TNTs in this cell type (Supplementary Fig. 3a, b).

## $H_2O_2$ generation by DUOX2 promotes lamellipodia formation and single-cell migration velocity

The presence of DUOX2 and $H_2O_2$ on dynamic, actin-associated structures suggest a role for DUOX2 in cytoskeletal rearrangements. Lamellipodia formation, a prominent feature of membrane extension, requires remodeling of the actin cytoskeleton. DUOX2 colocalized with actin at mature lamellipodia in H661 and BxPC3 cells (Fig. 4a, b; Supplementary Fig. 4a–c). DUOX2 was detected on early lamellipodia (Fig. 4c), a crosslinked "lattice" structure containing actin and cortactin. In accord, live cell video microscopy showed UnaG-DUOX2 trafficking to the plasma membrane during lamellipodium extension, with subsequent internalization once the extension ceased (Fig. 4d, Supplementary Movie 5), reinforcing the concept of dynamic DUOX2 on-off cycling to actin structures that require localized $H_2O_2$ generation.

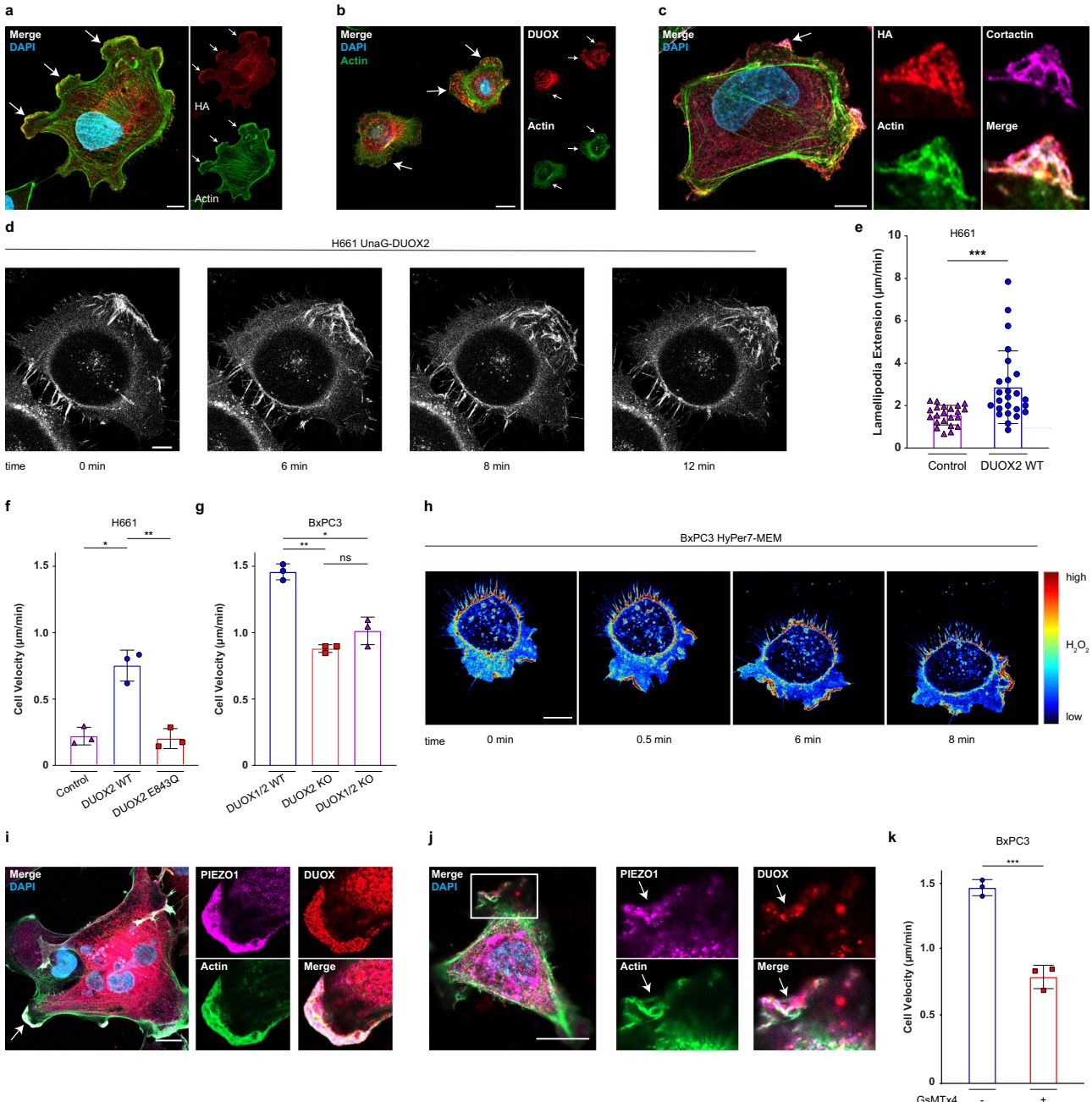

**Fig. 4 | H₂O₂ generation by DUOX2 promotes lamellipodia formation and single-cell migration velocity.** Localization of DUOX2 to extending lamellipodia in (**a**), DUOX2 WT expressing H661 cells (DUOX2 (HA, red); multiple lamellipodia denoted by white arrows) and in (**b**), BxPC3 DUOX1/2 WT cells (DUOX, red), inserts show spilt channels DUOX (red, top) and actin (green, bottom). **c** DUOX2 (HA, red) localized to a nascent lamellipodia, white arrow, colocalizing with cortactin (purple) and actin (green) in a typified lattice structure, inserts show split channel digital zoom of lamellipodia formation. **d** Localization of UnaG-DUOX2 (white) during lamellipodia formation; frames extracted from live cell confocal video microscopy at indicated time points (Supplementary Movie 5). **e** Kymograph analysis of lamellipodia protrusion velocity in control and DUOX2 WT expressing H661 cells, each point represents a single lamellipodia from separate cells across 3 independent experiments, Welch's *t* test, ***P < 0.005. Single cell migration velocity in **f**, H661 and **g**, BxPC3 cell lines, n = each point represents the average from 3

independent experiments with 25 cells tracked per experiment; Brown-Forsythe and Welch ANOVA with Dunnett's T3 post hoc test *P < 0.05, **P < 0.005. **h** H₂O₂ generation at the leading edge and rear retraction zone in migrating DUOX1/2 WT BxPC3 cells expressing HyPer7-MEM. Still images extracted from live cell video microscopy (Supplementary Movie 6). **i** PIEZO1 (purple) staining in H661 DUOX2 WT cells, colocalized with DUOX2 (red) at the leading edge of a lamellipodia, co-stained with actin (green), digital zoom of lamellipodia at white arrow highlighting colocalization. **j** Colocalization of DUOX (red) and PIEZO1 (purple) in DUOX1/2 WT BxPC3 cells; digital zoom of area denoted by white box highlighting colocalization at protrusions. **k** Single cell migration velocity in DUOX1/2 WT BxPC3 cells ± GsMTx4 (5 μM), n = each point represents the average from 3 independent experiments with 25 cells tracked per experiment, Welch's *t* test, ***P < 0.005. **e**–**g**, **k** Data are presented as mean ± SD; **a**–**c**, **i**, **j**, co-stained with DAPI (blue). **a**–**d**, **h**–**j** Scale bars 10 μm, microscopy analyses representative of n = 5.

Kymograph analysis indicated a significantly higher extension velocity in H661 cells expressing DUOX2 WT than in the control cell line (Fig. 4e).

Membrane protrusions are the primary mode of leading-edge cell migration[35]. Chemotaxis assays revealed that loss of DUOX2 activity significantly decreased migration velocity in both H661 and BxPC3 cell lines (Fig. 4f, g). Migration velocity of cells expressing catalytically inactive DUOX2 E843Q was comparable to H661 cells solely expressing DUOXA2 (control), implicating $H_2O_2$ generation by DUOX2 as a driver of directed migration (Fig. 4f). In BxPC3 cells the ablation of DUOX2 or of both DUOX enzymes (DUOX1/2 KO) decreased migration velocity to a similar extent (Fig. 4g), indicating that DUOX2 but not DUOX1 is contributing to directed migration in BxPC3 cells. $H_2O_2$ generation at the leading edge of migrating cells was detected in HyPer7-MEM expressing BxPC3 DUOX1/2 WT cells (Fig. 4h, Supplementary Movie 6). Without adding any extracellular stimuli a high intensity HyPer7 signal was observed at plasma membrane protrusions, reminiscent of the DUOX2 localization in fixed cells (see Fig. 4b, c). Lamellipodia extension occurred simultaneously with a flash of $H_2O_2$ at the plasma membrane. The time frame of these oxidized HyPer7-MEM signals was comparable to the duration of UnaG-DUOX2 localizing to the plasma membrane (1–2 min), supporting that these bursts of $H_2O_2$ are generated by DUOX2. We also observed the emergence of oxidized HyPer7-MEM signals at the rear of migrating cells. Rapid calcium entry at the front and rear of migrating cells is required for cell movement and is also required for DUOX2 activation[36,37]. The calcium channel PIEZO1 is expressed in many cell types, translating mechanical cues that promote cell spreading, focal adhesion formation, single cell migration and collective migration[38–40]. PIEZO1 is enriched at zones of extension, wound edges, and at the rear retraction zone of single migrating keratinocytes[39]. In H661 DUOX2 WT and DUOX2 expressing BxPC3 cells PIEZO1 colocalized with DUOX2 at prominent protrusions (Fig. 4i, j; Supplementary Fig. 4d). Inhibition of mechanosensitive channels such as PIEZO1 with GsMTx4 decreased cell migration velocity in BxPC3 DUOX1/2 cells to a similar extent as in DUOX2 knockout cells (Fig. 4k). Thus, PIEZO1 might be linked to calcium entry-induced activation of DUOX2 and subsequent $H_2O_2$ generation in dynamic cytoskeletal processes.

### $H_2O_2$ activated FER tyrosine kinase associates with DUOX2 to augment actin polymerization

Rapid, localized bursts of $H_2O_2$ facilitate spatially localized redox signaling that can trigger tyrosine kinase activation associated with actin dynamics. $H_2O_2$-mediated activation of FER tyrosine kinase has been directly linked to tyrosine phosphorylation of cortactin (Y421, Y466) and cell migration[41]. Immunofluorescence staining indicated marked colocalization of FER, DUOX2 and HyPer7-MEM on vesicles localized along a membrane protrusion (Fig. 5a; inserts: vesicles at white arrows). In both cell types, H661 DUOX2 WT cells (Fig. 5b) and BxPC3 DUOX1/2 WT cells (Supplementary Fig. 5a), DUOX2 and FER colocalized at the plasma membrane. Indeed, FER and DUOX2 colocalized also in both TNTs and apical circular protrusions (Supplementary Fig. 5b, c). DUOX2/FER colocalization at plasma membrane extensions was accompanied by high intensity oxidized HyPer7-MEM signals (Fig. 5c), setting the stage for localized oxidation and activation of FER. Given the close relationship between DUOX2 and FER, the newly developed FER inhibitor E260 was used[42]. Inhibition of FER resulted in a decrease in single-cell migration velocity (Fig. 5d). SiRNA knockdown of FER, confirmed by immunoblot and immunofluorescence (Supplementary Fig. 5d, e), also significantly decreased migration velocity in single cells (Supplementary Fig. 5f).

The pronounced colocalization of DUOX2 and FER, observed in two cell types and with several antibodies, suggested localized complex formation between these two proteins. Immunoprecipitation of FER from control or DUOX2 WT expressing H661 cell lysates detected FER-associated DUOX2, using both anti-HA and anti-DUOX2 antibodies, only in lysates containing DUOX2 WT (Fig. 5e). Moderate activation of DUOX2 by thapsigargin alone increased tyrosine phosphorylation of FER in H661 cells (Fig. 5f; $P = 0.0929$) and BxPC3 cells (Supplementary Fig. 5g, $P = 0.0549$), but considering the locally restricted HyPer7-MEM signal intensity at sites of DUOX2/FER interaction, whole cell lysates cannot reflect adequately this dynamic situation. Cortactin, a FER phosphorylation target colocalized with DUOX2 (see Fig. 2a, c, m, Fig. 4c), and thus cortactin phosphorylation at sites of DUOX2/FER localization was evaluated. Indeed, we detected phosphorylated cortactin (Y421) together with DUOX2 and an oxidized HyPer7-MEM signal at the leading edge of H661 DUOX2 WT cells and in TNTs (Fig. 5g, h; Supplementary Fig. 5h–j). Prominent plasma membrane localization of FER, phospho-cortactin and HyPer7-MEM was evident in both H661 cells and BxPC3 cells. A strong increase in pCortactin levels was observed in H661 DUOX2 WT cells and BxPC3 DUOX1/2 WT cells (Supplementary Fig. 5k). Examination of pCortactin intensities at membrane protrusions revealed a significant increase in cortactin phosphorylation in the presence of active DUOX2 (Supplementary Fig. 5l, m). These results suggest strongly that a DUOX2/FER/pCortactin pathway is driving actin polymerization. Indeed, polymerized F-actin assessed by quantifying the G/F actin ratio, was increased for both BxPC3 cells and H661 cells when DUOX2 was present (Fig. 5i, j), indicating that DUOX2 activity augments the rate of actin polymerization required for dynamic cytoskeletal remodeling.

### DUOX2 activity is essential for the retraction wave during epithelial wound healing

DUOX enzymes are expressed in barrier epithelia that encounter mechanical injury, trauma or infections, and require efficient wound healing mechanisms for protection. To probe the role of DUOX2 in regulating collective cell migration, several monolayer migration assays were carried out including barrier assays, wounding (scratch) assays, and in situ scratch assays for HyPer7-MEM quantification (Fig. 6a). Only BxPC3 cell lines were used for collective migration as H661 cells were not suitable due to their propensity for single cell migration. In IBIDI barrier inserts, reflecting self-propelled lamellipodia-driven collective crawling to close the gap, no significant differences in cell front migration speed or wound closure dynamics were detected when comparing BxPC3 cells expressing DUOX (DUOX1/2 WT) or not (DUOX2 KO, DUOX1/2 KO) (Fig. 6b).

In contrast, the dynamics of wound closure after mechanical disruption of confluent monolayers differed substantially when DUOX2 was present. In BxPC3 cells expressing DUOX1/2 WT the cell front speed displayed two highly elevated peaks at 3 h (Peak1) and 12 h (Peak2) with a sharp decline at 9 h, while the absence of DUOX2 significantly decreased cell front migration speed (Fig. 6c). The outcome of this DUOX2-dependent two-step process was a complete change in wound closure dynamics as the wound area initially increased 50% above the initial wounding area before the cell front reversed direction and collective movement for closure began (Supplementary Fig. 6a). Live cell imaging showed that the cell front of wounded DUOX1/2 expressing BxPC3 cells initially retracted, paused its motion briefly at ~9 h, and then reversed direction to close ~22-26 h after wounding (Fig. 6d, Supplementary Movie 7). In contrast, the cell front of BxPC3 cells lacking DUOX (DUOX2 KO, DUOX1/2 KO) started the inwards movement immediately after wounding, reaching full closure between 16 and 20 h (Fig. 6d; green line initial wound, red line wound edge at time point). To confirm the requirement for DUOX2 catalytic activity in this process, scratch assays were performed on BxPC3 DUOX1/2 WT cells in the presence of the calcium chelator BAPTA-AM and GKT137831, a non-specific NADPH oxidase inhibitor[43,44]. Both treatments abolished the retraction wave and decreased cell front speed (Fig. 6e). Ratiometric quantification of the HyPer7-MEM signal in BxPC3 DUOX1/2 WT cells during wound closure showed an increase in

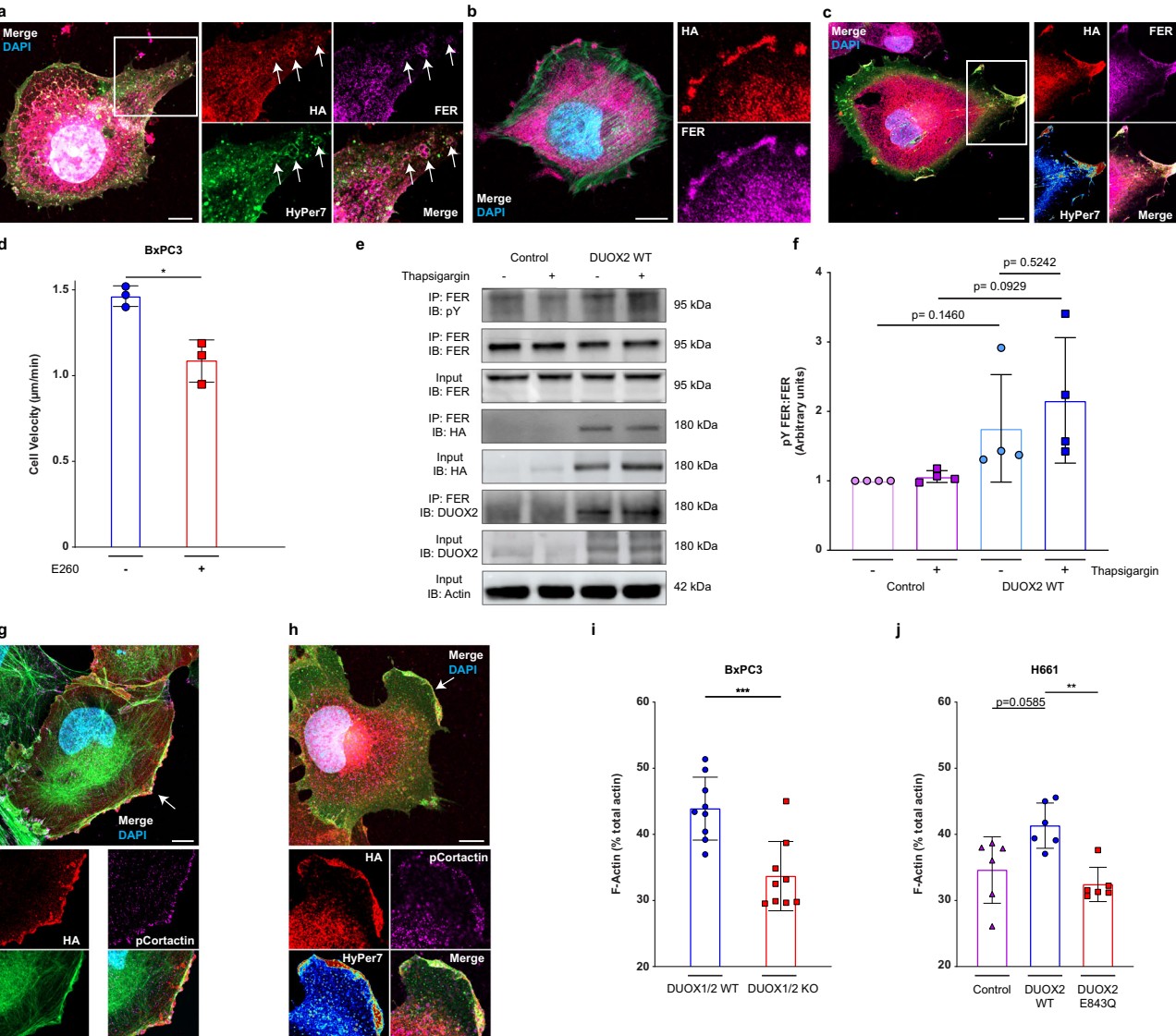

**Fig. 5 | H₂O₂ inducible FER tyrosine kinase is activated by DUOX2 to augment actin polymerization. a** Colocalization of DUOX2 (HA, red), FER (purple) and HyPer7-MEM (green) on intracellular vesicles in H661 DUOX2 WT cells; insert shows digital zoom with vesicles at white arrows. **b** DUOX2 (HA, red) and FER (purple) plasma membrane colocalization in DUOX2 WT expressing H661 cells. **c** Plasma membrane colocalization of DUOX2 (HA, red) and FER (purple) with HyPer7 pixel intensity heatmap in H661 DUOX2 WT cells. **d** Single cell migration velocity in DUOX1/2 WT BxPC3 cells ± FER inhibition (E260 5 μM), n = each point represents the average from 3 independent experiments with 25 cells tracked per experiment, Welch's t test, *P < 0.05. **e** Immunoprecipitation (IP) of FER from H661 cell lysates to assess both pY-FER and FER-DUOX2 interaction with and without DUOX2 stimulation (thapsigargin 1 μM, 1 h). Antibodies used for immunoblotting (IB) are indicated. **f** Densitometry analysis of pY FER from FER IPs in

H661 cells, each point represents a single replicate, each point represents 4 separate analyses, Welch's t test, p values displayed on graph. **g** DUOX2 (HA, red) and phospho-Cortactin (purple) membrane colocalization in DUOX2 WT expressing H661 cells. **h** Colocalization of DUOX2 (HA, red) and phospho-Cortactin (purple) in lamellipodia (white arrow) at sites of active H₂O₂ generation (HyPer7 signal). Quantification of F-Actin using G-Actin/F-Actin determination in BxPC3 (**i**) and H661 (**j**) cell lines as indicated; each point represents one technical replicate from 3 independent experiments. **i** Welch's t test, ***P < 0.0005, **j** Brown-Forsythe and Welch ANOVA with Dunnett's T3 post hoc test, **P < 0.005. **d, f, i, j** Data are presented as mean ± SD, **a, c, h**, HyPer7 presented as green for merged image and as heatmap (HyPer7 pixel intensity) in inserts to visualize H₂O₂ peaks. Scale bars 10 μm, microscopy analyses representative of n = 5.

H₂O₂ generation after wounding (Supplementary Fig. 6b, c; Supplementary Movie 8; recorded to 3 h) and during the retraction wave (Supplementary Fig. 6d, e; Supplementary Movie 8; recorded to 9 h). Adding exogenous H₂O₂ to BxPC3 DUOX1/2 KO cells increased cell front speed and delayed but partially restored the retraction wave (Fig. 6f, g; red line). Treatment of BxPC3 DUOX1/2 WT cells with H₂O₂ decreased speed and delayed the retraction wave (Fig. 6f, g; purple line). These effects are likely due to exogenous H₂O₂ acting globally on the monolayer, while mechanical wounding induces only spatio-temporal H₂O₂ production in the cells at the wound margins.

Visualization of DUOX2 location and activity in fixed cells at early time points after wounding showed recruitment to the wound edge up to the 3 h timepoint (Fig. 6h). In the same timeframe HyPer7-MEM 488 nm pixel intensity increased at the wound edge (Fig. 6h). DUOX2 activation requires calcium, likely provided by PIEZO1 channel opening. Calcium influx after mechanical wounding was visualized using the membrane permeable Fluo4-AM dye, showing a rapid calcium wave in the initial seconds post wounding (Supplementary Fig. 6f, g, Supplementary Movie 9). Further examination of Fluo4AM staining in live cells showed calcium containing vesicles in transit to leading-edge

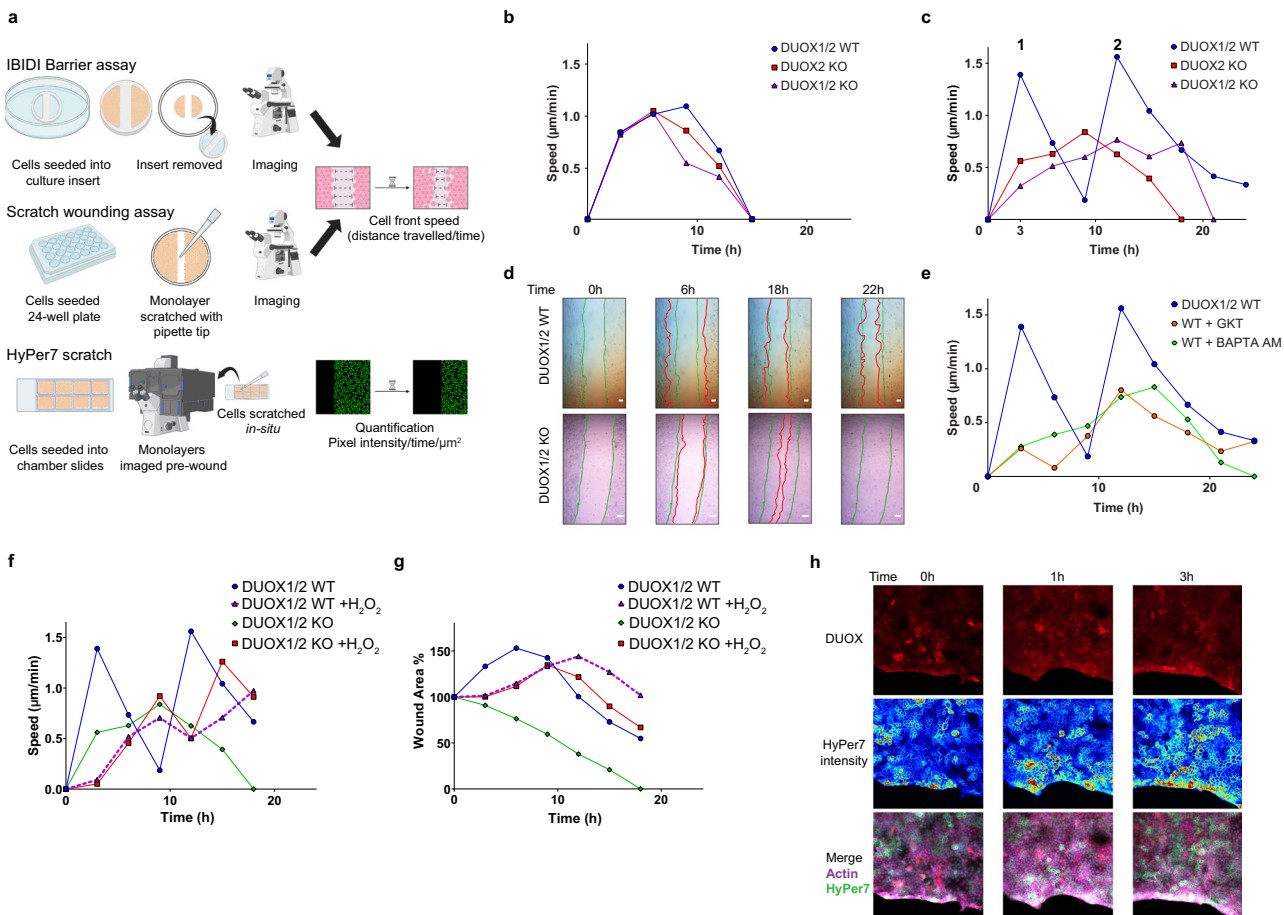

**Fig. 6 | DUOX2 activity is essential for the retraction wave during epithelial wound healing. a** Experimental design for wound healing analysis. **b** Cell front speed during barrier assay (IBIDI inserts) by BxPC3 DUOX1/2 WT (blue), DUOX2 KO (red), and DUOX1/2 KO (purple) cell lines, observed for 24 h. **c** Cell front speed during scratch assays in BxPC3 DUOX1/2 WT (blue), DUOX2 KO (red), and DUOX1/2 KO (purple) cell lines, observed for 24 h, numbers indicate peaks in BxPC3 DUOX1/2 WT cells that represent cell front retraction (1) and cell front extension (2). **d** Individual frames extracted from wound healing movies in BxPC3 DUOX1/2 WT cells (upper panels) and BxPC3 DUOX1/2 KO cells (lower panels) depicting differences in wound closure dynamics. Green lines represent wound edges at t = 0 h, red lines represent wound edges at indicated timepoints (Supplementary Movie 7) **e** Cell front speed in scratch assays in BxPC3 DUOX1/2 WT cells either

untreated (blue) or in the presence of NOX/DUOX inhibitor GKT137831 (10 μM, orange) or the cell permeant calcium chelator BAPTA-AM (20 μM, green). Cell front speed (**f**) and change in wound width (% of initial wound) (**g**) in BxPC3 DUOX1/2 WT or DUOX1/2 KO cell lines ± addition of exogenous $H_2O_2$ (25 μM) in scratch assays. **h** Fixed cell immunofluorescent staining of BxPC3 DUOX1/2 WT HyPer7-MEM cells, showing DUOX (red), HyPer7 (heatmap), and actin (purple), coverslips were fixed at the indicated timepoints post wounding. **b, c, e, f, g,** n = each point represents the average speed or wound area every 3 h, from 3 independent experiments, each with 6 replicates. **d, h** Scale bars 100 μm, microscopy analyses representative of n = 5. **a** Created in BioRender. Knaus, U. (2025) https://BioRender.com/1rsz4wv, modified in Adobe Illustrator.

membranes (Supplementary Fig. 6h), and increased Fluo4-AM intensities preceding both retraction and extension of the plasma membrane in single cells (Supplementary Movie 9). In summary, DUOX2-mediated $H_2O_2$ generation after mechanical wounding of epithelial cell layers induced a retraction wave with increased migration speed, resulting overall in slower wound healing.

## The cell front retraction wave requires both FER kinase and the mechanosensor PIEZO1

The association with DUOX2 and its redox-mediated activation points to FER tyrosine kinase as a potential downstream regulator of the cell front retraction wave. Indeed, FER colocalized with HyPer7-MEM at the wound edge (Fig. 7a), and inhibition of FER activity decreased cell front speed and abolished the retraction wave in DUOX1/2 WT expressing BxPC3 cells (Fig. 7b, c; purple line). The FER inhibitor did not substantially alter cell front speed or wound closure in DUOX1/2 KO cells (Fig. 7b, c; red line). Localized collective retraction after mechanically induced wounding of a keratinocyte layer has been recently connected to PIEZO1 activation[39]. Enrichment of PIEZO1 at the retracting wound

edge was observed for up to 3 h, accompanied by FER and DUOX2 localization to the same areas (Fig. 7d, e). Treatment with the PIEZO1 agonist Yoda1 decreased BxPC3 DUOX1/2 WT cell front speed, likely due to global versus spatiotemporal activation, but did not impede the retraction wave (Fig. 7f, dark blue line). Adding Yoda1 to DUOX1/2 KO cells did not create a pronounced retraction wave (Fig. 7f, red line), but we noticed that cells in some areas of the wound edge attempted to retract but could not progress, likely as DUOX2-generated $H_2O_2$ was not present (Fig. 7g, red arrows; Supplementary Movie 10). Inhibition of PIEZO1 by GsMTx4 decreased cell front speed substantially (Fig. 7f, light blue line) and abolished the retraction wave (Fig. 7h) in DUOX1/2 WT cells. In the absence of DUOX2 GsMTx4 treatment did not substantially alter the behavior of DUOX1/2 KO cells (Fig. 7f, pink line versus Fig. 7b, green line).

Thus, after mechanical disruption of an epithelial layer the combined activity of a PIEZO1-DUOX2-FER axis in the early stages of cell movement led to slower wound closure. Rapid re-epithelization of wounds is considered beneficial, yet this complex cell behavior points rather to a physiological advantage. Comparison of cell front closure

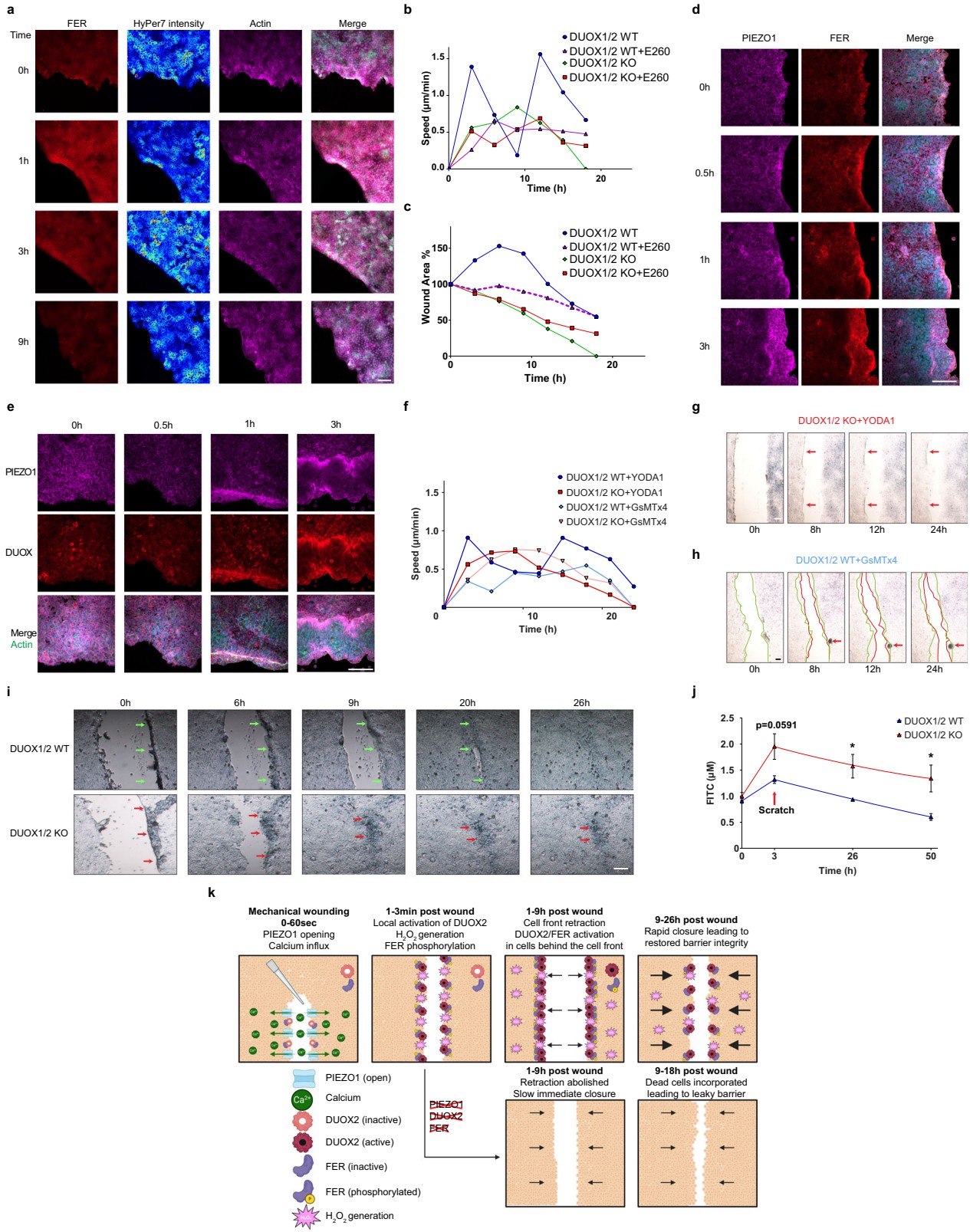

between DUOX1/2 WT and DUOX1/2 KO BxPC3 cell lines revealed a distinct difference in removal of wound edge cells damaged by mechanical wounding. In DUOX1/2 WT cells, retraction of the cell front occurred until viable cells formed the cell front, and necrotic/apoptotic cells and cell debris were removed (Fig. 7i). Damaged cells were absent after the slower wound closure was achieved (Fig. 7i; green arrows). In contrast, when the retraction wave cannot occur due to the

absence of DUOX2, the damaged cell front was not removed but became incorporated into the monolayer after wound closure (DUOX1/2 KO, Fig. 7i; red arrows). This failure to clean up the wound margin before closure was also evident when PIEZO1 was inhibited in DUOX1/2 WT cells (Fig. 7h; red arrow). We reasoned that incorporated cell debris and dead cells will increase the permeability of epithelial layers. As anticipated, DUOX1/2 KO cell layers (red line) were more

**Fig. 7 | Both FER kinase and PIEZO1 are required for the cell front retraction wave. a** FER (red) localization during wound healing in BxPC3 DUOX1/2 WT HyPer7-MEM cells, HyPer7 (merge, green; or intensity heatmap) and actin (purple), coverslips were fixed at the indicated timepoints post wounding. Cell front speed (**b**) and change in wound area (**c**) in BxPC3 DUOX1/2 and DUOX1/2 KO cell lines ± FER inhibition (E260 20 μM). **d** Colocalization of PIEZO1 (purple) and FER (red) at the wound edge during scratch assay in fixed BxPC3 DUOX1/2 WT cells. **e** Colocalization of PIEZO1 (purple) and DUOX (red) at the wound edge in fixed BxPC3 DUOX1/2 WT cells. **f** Cell front speed during PIEZO1 stimulation (Yoda1 5 μM) or calcium channel inhibition (GsMTx4 5 μM) in BxPC3 DUOX1/2 WT cells (blue, Yoda1; light blue, GsMTx4) or BxPC3 DUOX1/2 KO cells (red, Yoda1; light red, GsMTx4). Brightfield frames extracted from live cell scratch assay movies showing time-dependent wound closure for (**g**), Yoda1-treated BxPC3 DUOX1/2 KO cells, red arrows indicate areas of cell front arrest (Supplementary Movie 10), and **h**, GsMTx4-treated BxPC3 DUOX1/2 WT cells, green lines indicate initial wound edges at t = 0 h,

red lines indicate wound edges at the indicated timepoints, red arrows show area of damaged cells. **i** Brightfield images extracted from live cell scratch assays at indicated timepoints, see elimination of damaged cell front in BxPC3 DUOX1/2 WT cells (upper panels, green arrows) and inefficient incorporation of the damaged cell front in BxPC3 DUOX1/2 KO cells (lower panels, red arrows). **j** Analysis of monolayer permeability using FITC-dextran (4 kDa) in BxPC3 DUOX1/2 WT (blue) and DUOX1/2 KO (red) cell lines for 24 h pre-wound followed by measurements at 3 h, 26 h, and 50 h post wounding, individual Welch's $t$ test at each time point, *$P < 0.05$. Each point represents the average from 3 independent experiments, each with 3 replicates, Data are presented as mean ± SEM. **k**, Scheme of PIEZO1-DUOX2-FER engagement during wound healing. **b**, **c**, **f**, $n$ = each point represents the average cell front speed or wound area for every 3 h from 3 independent experiments, each with 6 replicates. **a**, **d**, **e**, **g**, **h**, **i** Scale bars 100 μm, microscopy analyses representative of $n$ = 5. **k** Created in BioRender. Knaus, U. (2025) https://BioRender.com/putget1, modified in Adobe Illustrator.

permeable than DUOX1/2 WT cell layers (blue line) after wounding (Fig. 7j). While cell layers containing DUOX2 became even tighter 48 h post wounding, cell layers without DUOX2 expression did not return to the pre-wounding state. Thus, the initial cell front retraction improved efficient wound closure and epithelial barrier function, a scenario summarized schematically in Fig. 7k.

## Discussion

Dynamic cytoskeletal processes including actin filament assembly and disassembly, exploratory cell behaviors, cell-cell communication, front-rear polarization, and directed migration require rapidly changing signals. Redox signaling provides an ideal on/off switch for signal propagation to the regulatory network governing actin reorganization, but understanding how ROS signal specificity can be accomplished in a highly dynamic situation is unresolved. We conceptualized that membrane-associated delivery of a ROS generating enzyme to sites of dynamic actin remodeling, followed by in-situ activation with a brief increase of the $H_2O_2$ concentration, would provide targeted and reversible oxidation. Fluorescently tagged UnaG-DUOX2 revealed continuous trafficking of DUOX2 containing vesicles to areas of comprehensive actin remodeling, accompanied by bursts of $H_2O_2$ at sites of extension or retraction, followed by DUOX2 internalization and recycling. These DUOX2/$H_2O_2$ containing vesicles clustered not only in actively extending areas of lamellipodia, in the front and rear of migrating cells or in dorsal ruffles, but also at the tip of nascent TNTs and as cargo shuttled between two cells via TNT. Intercellular transfer of active DUOX2 within membrane bulges that contained vesicles, actin, acetylated α-tubulin and motor proteins is reminiscent of the transport of mitochondria via TNTs in neuronal cells[31].

These cytoskeletal events have in common the need for rapid, localized actin polymerization. Cells lacking active DUOX2 were still capable of migration and forming TNTs and lamellipodia, likely due to other $H_2O_2$ producing enzymes (e.g., LOX, LOXL2), but DUOX2/$H_2O_2$ significantly elevated polymerized F-actin content and markedly increased the rate and speed at which cells carried out these processes. DUOX2 as a calcium-activated NADPH oxidase is perfectly suited as conduit for locally released calcium pulses that regulate cycles of local cell edge expansion, adhesion and retraction[45,46]. These cytoskeletal rearrangements are essential for cell migration, yet extracellular stimuli, the cell type involved, and the mode of migration strongly influences which signaling networks will be activated. Collective movement to fill a gap did not require DUOX2 activity in BxPC3 cells, yet DUOX2 was essential to mediate the rapid retraction wave and the subsequent peak of accelerated wound closure. Such efficient healing response after injury in conjunction with barrier restoration is vital in the intestinal epithelium where DUOX2 is prominently expressed at the apical surface. In the lung epithelium, both DUOX2 and DUOX1 are present and could generate $H_2O_2$ at wound margins. In fact, Wesley

and colleagues reported wounding-induced, ATP-stimulated $H_2O_2$ production by DUOX1 accelerated wound closure in bronchial epithelial cells[47]. DUOX-generated $H_2O_2$ has also been connected to wound healing and long-range signaling in keratinocytes, zebrafish and *Drosophila*. In certain cell types including cancer cells $H_2O_2$ producing NADPH oxidases such as DUOX1/2 or NOX4 provide redox signaling for cytoskeletal dynamics, enhanced migration and metastasis[12,13,48,49], while in other cell types NOX4 was dispensable and NOX1 promoted directed migration upon wounding[50,51]. The pancreatic BxPC3 cells used here express endogenously DUOX2 and DUOX1, but DUOX1 did not contribute significantly to cytoskeletal dynamics. Many cell types express several NADPH oxidases simultaneously, but deeper understanding of functional ROS signaling specificity awaits development of novel tools for tracking different oxidase isoforms in space and time, and of strictly isoform specific oxidase inhibitors.

In migrating cells, a spatial gradient of endoplasmic reticulum and plasma membrane contacts defines front-rear polarization, localized tyrosine kinase and phosphatase signaling, and migration speed[52]. Membrane-attached HyPer7 live cell imaging visualized short $H_2O_2$ flashes at the leading edge of migrating cells, while $H_2O_2$ generation was more sustained at the trailing edge. Vesicle-associated $H_2O_2$ will alter the activation status of tyrosine kinases, phosphatases and cytoskeleton-associated thiol-containing proteins by reversible oxidation. Regarding non-receptor tyrosine kinases, redox-dependent SRC kinase activation has been linked to cell motility[13,53]. Here, we show DUOX2 co-localizing with the non-receptor tyrosine kinase FER in $H_2O_2$ generating transport vesicles, at the plasma membrane, in TNT cargo and at the leading edge of migrating cells and wounded cell margins. DUOX2-FER association and increased pTyr incorporation in FER supported the $H_2O_2$-mediated activation of FER and phosphorylation of its downstream target cortactin[41]. $H_2O_2$ was also generated in DUOX2 containing vesicles at the rear of migrating cells. The oxidation targets at the rear are likely connected to myosin activation, actomyosin contraction and trailing edge retraction (e.g., oxidation-activated RhoA GTPase)[54].

Wound closure requires a balance between inflammation, infection control, and re-epithelialization. DUOX2 activity markedly altered the process of wound closure, generating a retraction wave that will facilitate efferocytosis and microbe removal, thereby ensuring restoration of barrier function. It is important to note that DUOX2 is predominantly expressed at the apical side of physical barrier tissues including intestinal and lung epithelia, and in epidermal keratinocytes, and thus is perfectly positioned to regulate wound healing processes. DUOX2-induced redox signaling was essential for directionality and speed of the collective backward-stop-forward cell movement after mechanical disruption of the epithelial layer. This is reminiscent of the involvement of the mechanosensor PIEZO1 in early wound margin retraction in keratinocytes and *Drosophila* embryos[39,55]. Ablation of

*Piezo* decreased calcium influx, damage-induced ROS and macrophage recruitment, leading to faster but ineffective wound closure and higher mortality of wounded embryos[55]. We show here that in mechanically wounded epithelial layers a DUOX2-FER pathway transmitted the PIEZO1 signal to induce an accelerated, synchronized retraction wave. Loss of the retraction wave increased permeability of the inadequately sealed cell layer, leaving the epithelium susceptible even after wound closure. This carries implications for inflammatory bowel disease patients with *DUOX2* loss-of-function variants as the intestinal barrier will be vulnerable to transfer of microbes and microbial products, resulting in chronic inflammation[56]. From a general perspective, the ability of cells to transport vesicle-incorporated enzymes for short, localized bursts of $H_2O_2$ to sites of active actin remodeling in processes connected to sensing, cell-cell communication and motility points to a fundamental role of redox signaling in cytoskeletal dynamics.

## Methods

### Reagents

The following reagents were used: BAPTA-AM (196419), diphenyle-neiodonium chloride (DPI) (300260), Calbiochem; phorbol 12-myristate 13-acetate (PMA) (P8139), thapsigargin (T9033), LPS *E.coli* (L2630), homovanillic acid (306-08-1), N-ethylmaleimide (04260-5G-F), hydrogen peroxide (H1009-5mL), and epidermal growth factor (E4127-1MG) all Sigma Aldrich; BODIPY FL phallacidin (B607), BODIPY 558/568 phalloidin (B3475), DAPI (D-1306) and Fluo4-AM (F14201) by Invitrogen; recombinant human IFNγ (Biolegend, 570202); FITC-dextran 4 kDa (TdB Labs, 60842-46-8); GKT137831 (DC Chemicals, DC8118); Yoda1 (HY-18723), GsMTx4 (HY-P1410), E260 (HY-112097), all MedChemExpress. SPY555 FastAct (CY-SC2-5) from Cytoskeleton Inc. ON-TARGETplus Non-targeting siRNA #1 (D-001810-01-05), ON-TARGETplus Human FER (2241) siRNA SMARTpool (L-003129-00-0005), and DharmaFECT 2 Transfection Reagent (T-2002-01), all from Horizon Discovery Biosciences.

### Antibodies

Primary antibodies used: mouse monoclonal anti-Actin (Cytoskeleton, AAN02-S) 1:1000, rabbit polyclonal anti-F-Actin (Sigma Aldrich, A0266) 1:4000, mouse monoclonal anti-acetylated Alpha tubulin (Sigma Aldrich, T7451) 1:100, rabbit monoclonal anti-EEA1 (Cell Signaling Technology 3288S) 1:100, mouse monoclonal anti-LAMP1 (DSHB H4A3-c) 1:100, rabbit monoclonal anti-GAP43 (Invitrogen MA5-32256) 1:100, rabbit monoclonal anti-Beta tubulin (Epitomics, EP1331Y) 1:1000, mouse anti-phosphotyrosine (Upstate, 05-321) 1:500, rabbit anti-Cortactin (Epitomics, EP1922Y) 1:100, rabbit polyclonal anti-phospho-Cortactin (Tyr421) (Invitrogen, 44-854 G) 1:100, mouse monoclonal anti-FER (Invitrogen, MA5-15357) IF 1:100, IB 1:500, rabbit polyclonal anti-FER (Proteintech, 25287-I-AP) IF 1:100, IP 2 μg, mouse monoclonal anti-PIEZO1 (Invitrogen, MA5-32876) 1:100, rabbit poly-clonal anti-Myo10 (Invitrogen, PA5-55019) 1:100, mouse anti-RAB11 (BD Transduction, 610656) 1:100, rabbit polyclonal anti-SNX4 (Synaptic Systems 392 003) 1:100, rabbit polyclonal anti-ARL13B (Proteintech, 17711-1-AP) 1:100, rabbit polyclonal anti-CEP164 (Proteintech, 16851765) 1:800, mouse monoclonal anti-HA tag (Covance, MMS-101P) IF 1:100, IB 1:1000, rabbit anti-DUOX custom rabbit antibody #7936[57] IF 1:100, rabbit anti-DUOX2 (custom rabbit antibody #7959[22] IB 1:1000). For antibodies see also Supplementary Data 1.

Secondary antibodies: goat anti-mouse IgG-HRP (Southern Biotech, 1030-05) 1:10,000, goat anti-rabbit IgG-HRP (Southern Biotech, 4030-05) 1:10,000; goat anti-rabbit IgG (H + L) AlexaFluor 488 con-jugated (A11008), goat anti-rabbit IgG (H + L) AlexaFluor 546 con-jugated (A11035), goat anti-rabbit IgG (H + L) AlexaFluor 594 conjugated (A11012), goat anti-rabbit IgG (H + L) AlexaFluor 647 con-jugated (A21244), goat anti-mouse IgG (H + L) AlexaFluor 488 con-jugated (A11029), goat anti-mouse IgG (H + L) AlexaFluor 546 conjugated (A11030), goat anti-mouse IgG (H + L) AlexaFluor 647

conjugated (A21236), goat anti-mouse IgG phycoerythrin conjugated (M30004-1) (all Invitrogen, 1:1000).

### Plasmids

Plasmids in pcDNA3.1: HA-DUOX2[28]; DUOXA2[25]; DUOX2 WT denotes human DUOX2 tagged between aa27-28 with HA-HA and Strep (St) tags, constructed by subcloning a 188-bp synthetic DNA (Eurofins) with BamH1 and BstX1 sites; DUOX2 E843Q point mutation was introduced into DUOX2 WT; UnaG-DUOX2: Codon optimized UnaG sequence[26] was synthesized (Eurofins MWG) with in-frame restriction sites (BamH1-BstX1) and inserted with linkers into DUOX2 between aa27-28. pCMV-HyPer7-MEM was a gift from Vsevolod Belousov (Addgene plasmid # 136465; http://n2t.net/addgene:136465;RRID:Addgene_136465); SypHer7-MEM C121S mutant was generated by mutagenesis. Lentivirus: DUOX2 and DUOXA2 were cloned into CGW lentiviral expression vectors[25]. Bicistronic vectors were used for DUOX2 (WT, E843Q) and DUOXA2 co-expressing CFP or mCherry to facilitate stepwise flow cytometry sorting.

### Cell Culture and transfection

NCI-H661 cells (ATCC HTB-183) lacking NOX/DUOX enzymes[22] and BxPC3 cells (ATCC CRL-1687) which express mainly *DUOX2* (mean Ct-24) were cultivated at 37 °C and 5% $CO_2$ in RPMI-1640 medium with GlutaMAX (Gibco 61870036), 10% heat inactivated fetal bovine serum (FBS; F7524, ThermoFisher Scientific) and 1% sodium pyruvate (11360070, ThermoFisher Scientific). HEK293FT cells were cultured at 37 °C and 5% $CO_2$ in DMEM high glucose (11965092, ThermoFisher Scientific) with 10% FBS (Supplementary Data 1). NCI-H661 cells were transiently transfected using FuGene HD (E2311, Promega) as per manufacturer's protocol, assays were carried out 48 h post transfec-tion. All DUOX2 constructs were co-transfected with DUOXA2. Lenti-viral transduction: virus particles were generated in HEK293FT cells as described elsewhere[58]. Viral particle concentration was carried out using PEG-it virus precipitation solution (LV810A-1, Cambridge Bios-ciences) as per manufacturer's protocol. BxPC3 cells and NCI-H661 cells were incubated with concentrated virus for 24 h in the presence of polybrene (4 μg/ml; Sigma). NCI-H661 cells were transduced in a two-step process generating DUOXA2 expressing cells first, sorted by mCherry, followed by transduction of either DUOX2 WT or DUOX2 E843Q, sorted by CFP. HyPer7-MEM was transduced into both DUOX2 WT/DUOXA2 and DUOXA2 only (Control) H661 cell lines, and into BxPC3 DUOX1/2 WT cells. HyPer7-MEM expressing cells were sorted by HyPer7 488 nm fluorescence in the presence of 10 μM $H_2O_2$. The fol-lowing stable expressing cell lines were generated: NCI-H661 DUOXA2, NCI-H661 DUOXA2/DUOX2 WT, NCI-H661 DUOXA2/DUOX2 E843Q, NCI-H661 DUOXA2/HyPer7-MEM, NCI-H661 DUOXA2/DUOX2 WT/HyPer7-MEM, BxPC3 DUOX1/2 WT HyPer7-MEM.

### CRISPR/Cas9-mediated ablation of DUOX2 and DUOX1

Knockout of DUOX2 and remaining low level DUOX1 from BxPC3 cells was performed by transfection with a guide RNA delivering pSpCas9 (BB)−2A-Puro (Addgene, 62988), which co-expresses the Cas9 nuclease and carries a puromycin selection marker (PuroR). Com-plementary oligonucleotides with BbsI overhang at both ends (forward strand: 5′-CACCGAAGTGCAGCGCTATGACGGC-3′; reverse strand: 5′-AAACGCCGTCATAGCGCTGCACTTC-3′) and 20 nucleotides of guide RNA targeting DUOX2 at exon 3 were synthesized by Eurofins (Ebers-berg). Similarly, DUOX1 was targeted at exon 4 (forward strand: 5′-CACCGAGCTCAGAACCCCATTTCGT-3′; reverse strand: 5′-AAAC ACGAAATGGGGTTCTGAGCTC-3′). Guide RNA-containing oligonu-cleotides were subcloned into the Cas9 co-expressing plasmid vector using phosphorylation by T4 polynucleotide kinase (M0201S, New England Biolabs), annealing, and ligation. Transfection of targeting plasmids into BxPC3 cells was performed using EZT-BXPC-1 (EZ Bio-systems) and 1 mg/ml of puromycin for three weeks to select targeted

colonies. Targeted BxPC3 cells were validated by PCR and DNA sequencing.

## SiRNA-mediated FER knockdown

BxPC3 DUOX2 WT cells were seeded at $3 \times 10^5$ cells per well in 12-well culture plates. 24 h after seeding cells were transfected using DharmaFECT 2 (Horizon T-2002) as per the manufacturer's protocol using three conditions: negative control siRNA, FER SmartPool siRNA, or untreated. Cells were washed 24 h post transfection, incubated for additional 48 h, and analyzed for FER knockdown 72 h post transfection in immunoblot and immunofluorescence. For migration analysis, cells were trypsinzed 48 h after siRNA transfection, counted, and reseeded into chemotaxis µ-slides (IBIDI 80326) and processed 24 h later for single cell migration velocity.

## H₂O₂ measurement

$H_2O_2$ generation by DUOX2, stimulated by addition of 100 ng/ml PMA and 1 µM thapsigargin for 1 h in PBS with calcium and magnesium, was measured using the homovanillic acid assay (HVA)[59]. $3 \times 10^5$ cells were seeded in a 12 well plate 24 h before the HVA assay was carried out. Fluorescence was determined using a Synergy MX plate reader (Biotek). $H_2O_2$ production was quantified using the $H_2O_2$ standard curve and cell lysate protein concentration, as measured by BCA assay.

## Immunoblotting and immunoprecipitation

Protein for immunoblotting was extracted in RIPA buffer (110 mM Tris-HCL pH 7.5, 140 mM NaCl, 2% NP-40, 2% sodium deoxycholate and 0.2% SDS) containing protease inhibitors (cOmplete Mini; Roche), separated by SDS-PAGE, blotted, and incubated with primary antibodies overnight, followed by secondary HRP-conjugated anti-rabbit (Southern Biotech) or anti-mouse antibodies (Cell Signaling, Danvers) for 1 h. Proteins were visualized using ECL reagent (Pierce, 32106). For immunoprecipitation, cells were lysed in TX-100 buffer (1% TX-100, 50 mM HEPES pH 7.35, 2.5 mM EDTA, 150 mM NaCl, 30 mM beta-glycerophosphate, 1 mM sodium orthovanadate, 1 mM PMSF, 12.5 mM NaF, 25 µM leupeptin, 12.5 µM pepstatin, and cOmplete MINI protease inhibitor tablet, in dH₂O). Cell lysates were incubated with 2 µg rabbit polyclonal anti-FER antibody (Proteintech) for 2 h at 4 °C with rocking. Protein A Sepharose beads (75 mg/mL, P3381, Sigma Aldrich) in PBS were added to the lysate in a 1:1 ratio and incubated overnight at 4 °C. After washes and centrifugation, beads were heated in 2X Laemmli buffer for 5 min at 90 °C and centrifuged (20,000 g for 10 min). Equal volumes of supernatant were separated by SDS-PAGE and immunoblotted as above. Samples were subsequently immunoblotted with mouse monoclonal anti-FER antibody (Invitrogen). Phospho-FER quantification was carried out by densitometry analysis using ImageJ with anti-FER immunoblots being used for normalization (n = 4). Blots presented in figures were cropped and aligned using ICY imaging software with no additional processing.

## HyPer7-MEM imaging and quantification

HyPer7-MEM expressing cells were seeded into 8 well µ-slides (80826, IBIDI) and imaged on an inverted Nikon Ti microscope with a Yokogawa spinning disk integrated by Andor Fusion, using a ×10 and ×20 objective. Microscope environmental controls were maintained at 37 °C and 5% CO₂. Cells were imaged for 30-60 min with the first 5 min discarded to allow for probe equalization. Indicated stimulants or inhibitors were added to the wells directly on the microscope stage while imaging was paused. The following control experiments were carried out: First, NCI-H661 HyPer7-MEM cells coexpressing DUOX2 WT/DUOXA2 or DUOXA2 alone (Control) were imaged without stimulation in RPMI-1640 media supplemented with 10% FBS and without phenol red, BAPTA-AM (20 µM) was added after 60 min and the cells were recorded for a further 30 min (frame rate: initial 60 min with 2 min intervals; 30 min post stimulation 4 min intervals). Second, NCI-

H661 DUOX2 WT/DUOXA2 cells transiently transfected with HyPer7-MEM were treated with either $H_2O_2$ (15 µM) or diphenyleneiodonium chloride (DPI) (15 µM) (frame rate: initial 60 min with 2 min intervals, post stimulation 2 h with 1 min intervals). Third, NCI-H661 DUOX2 WT/DUOXA2 cells and DUOXA2 (Control) cells were transiently transfected with either HyPer7-MEM or SypHer7-MEM; HyPer7 and SypHer7 ratio was measured for 60 min followed by $H_2O_2$ (15 µM) treatment and recorded for a further 40 min (frame rate initial 60 min with 2 min intervals, post $H_2O_2$ treatment with 30 s intervals). In all conditions HyPer7 fluorescence was excited at 405 nm and 488 nm and detected at 499–560 nm. 3 fields of view were imaged per well, with 2 wells per condition. Time series were analyzed using ICY Bioimage analysis software (Institut Pasteur). Background was subtracted from each channel and the 488 nm series was divided by the 405 nm series to produce the 488/405 nm HyPer7 ratio. A region of interest was placed around individual cells, and the mean pixel intensity value was obtained for the entire time series using the ROI Intensity Evolution plugin. 3–6 cells were measured per field of view, under the criteria that only single cells were included within a ROI. HyPer7 intensity from measured cells were combined to plot mean HyPer7 ratio/time for each condition. The HyPer7-MEM signal at membrane protrusions was quantified as above, except the ROI was placed specifically around areas of extending plasma membrane.

## Immunofluorescence

Fixed cell imaging: Cells were seeded on 13 mm coverslips (631-0148, VWR) placed in 24-well plates. For some localization studies cells were fixed 24 h after seeding or 48 h after transient transfection. TNT staining was carried out after additional procedures, 24 h serum starvation for increased formation of horizontal TNTs or adding mouse fibroblast conditioned media (mFCM; generated in laboratory) for 24 h, stimulating the formation of apical protrusions. Circular Dorsal Ruffle formation was stimulated by epidermal growth factor (10 µg/mL) for 5 min. For fixed wound healing analysis, BxPC3 cells were grown until 100% confluency was reached. Cell monolayers were scratched with a P10 pipette tip and fixed at the indicated time points. All fixation was carried out with 2% PFA for 15 min at room temperature. Cells expressing HyPer7-MEM were washed with N-Ethylmaleimide (50 mM) before fixation to preserve the redox state of HyPer7. 10 min permeabilization with saponin (0.5% in PBS) was followed by 1 h blocking (3% goat serum, 1% BSA and 3% glycine). Coverslips were then incubated with primary antibodies in 3% BSA, 0.1% saponin in PBS for 3 h at room temperature (RT) in darkness. Where actin staining was required, BODIPY FL phallacidin or BODIPY 558/568 phalloidin were included with the primary antibody solution (1:100). After washing, coverslips were incubated with Alexa Fluor secondary antibodies (3% BSA, 0.1% saponin in PBS) for 1 h at RT in darkness. Coverslips were stained with DAPI for 3 min, washed, and mounted on slides with Fluoromount-G (00-4958-02, ThermoFisher Scientific). Fixed cell imaging was carried out on a Zeiss LSM800 Airyscan confocal microscope with a Plan-Apochromat 63X oil immersion objective. Images were processed using ICY Bioimage analysis software.

Live cell imaging: Cells were seeded into 8-well µ-slides (80826, IBIDI). NCI-H661 DUOXA2 cells were transiently transfected with UnaG-DUOX2 in the IBIDI slide using FuGene HD and imaged after 48 h. HyPer7-MEM stable expressing cell lines were seeded directly into the wells. Cells were imaged 24 h after seeding or after treatment as described above for TNT imaging. Medium was exchanged to RPMI-1640 without phenol red (11835030, ThermoFisher Scientific) either serum free or supplemented with 10% FBS that contains bilirubin. Live cell imaging was carried out on an Olympus IX83 (FV3000) confocal microscope using an UPLFLN 60X/NA1.35 oil immersion objective. Microscope environmental controls were maintained at 37 °C and 5% CO₂. For live cell movies, HyPer7-MEM was imaged at $1024 \times 1024$

resolution at a frame rate of 10 s per image (frame 5.8 s, interval 4 s). UnaG fluorescence is sensitive to photo-oxidation and can bleach at higher laser power. To minimize bleaching, UnaG-DUOX2 was imaged on low laser power (<1% 488 nm) at a frame rate of ~20 s per image (frame 5.8 s, interval 15 s). Fluo4-AM live cell imaging was carried out as above with the addition of 1 h pre-incubation with 5 μM Fluo4-AM and 1X solution of the live actin stain SPY555-FastAct.

In fixed and live cell imaging of HyPer7-MEM where quantification was not carried out, images are displayed as the 488 nm channel. Images are either displayed in green pseudo color or converted to a heatmap. Heatmaps applied in ICY are based on pixel intensity values, with high values represented in red declining to low values in blue.

### Lamellipodia kymograph analysis

Time series used in HyPer7-MEM quantification were used for kymograph generation. NCI-H661 DUOX2 WT/DUOXA2 and DUOXA2 cells stably expressing HyPer7-MEM were chosen from the pre-stimulation time points to quantify protrusion velocity of lamellipodia. HyPer7 488 nm fluorescence was used to visualize lamellipodia formation. ImageJ was used to generate kymographs of lamellipodia from 20–25 individual cells, per condition, across 3 experiments.

### Single cell migration

BxPC3 cells (DUOX1/2 WT, DUOX2 KO, DUOX1/2 KO) and NCI-H661 cells (DUOXA2 control, DUOX2 WT/DUOXA2, DUOX2 E843Q/DUOXA2) cells were seeded into chemotaxis μ-slides (IBIDI 80326). 24 h after seeding, media was exchanged for RPMI-1640 without phenol red without FBS. RPMI-1640 media with 20% FBS was used as chemoattractant to stimulate directed cell migration. Cells were imaged for 6 h at 5 min intervals at 37 °C on a Nikon Ti microscope using a 10X objective. Individual cells were tracked using the ImageJ Manual Tracking plugin. 20–25 cells were tracked per replicate for each cell line over 3 independent experiments to calculate average migration velocity.

### TNT quantification

NCI-H661 cells (DUOXA2 control, DUOX2 WT/DUOXA2, DUOX2 E843Q/DUOXA2) seeded at $3 \times 10^5$ cells in 12-well cell culture plates were observed with a light microscope (Nikon TMS) after 24 h of serum starvation, 10 μM of GKT137831 was added to selected wells during 24 h serum starvation, with a reapplication after 10 h. 5 fields of view were observed for each well, with 3 wells per cell line, over 4 independent experiments. Cells possessing at least 1 TNT were expressed as a percentage of total cells within a field of view.

### Wound healing assays

Wound healing scratch assay: BxPC3 cells (DUOX1/2 WT, DUOX2 KO, DUOX1/2 KO) were seeded into 24-well cell culture plates and grown to confluency. Cell monolayers were scratched with a P10 pipette tip to create a linear wound. Wells were washed before the medium was exchanged for RPMI-1640 without phenol red, supplemented with 10% FBS. Monolayers were imaged with a Nikon Ti microscope, with environmental controls maintained at 37 °C. $CO_2$ was not used during wound healing assays, cell media was buffered with 10 mM HEPES to maintain pH. Images were taken every 30 min for 24 h using a 2X objective. The resulting time series were processed in ICY Bioimaging software. Automated cell front detection was inaccurate due to differences in contrast produced by the position of particular wells, shadows appeared on wells at plate edges while central wells were overly bright. Thus, cell front speeds were manually calculated. Distance was measured at 6 points along the full length of the wound for each time series. The average speed of the cell front was calculated using the distance traveled every 3 h (μm/3 h was converted to μm/min). Inhibitors or agonists were added to cell media 30-40 min before wounding, then added again to fresh RPMI-1640 medium without

phenol red before imaging commenced. Concentrations used: BAPTA-AM 20 μM (added to medium a second time at 6 h post wounding), $H_2O_2$ 25 μM, E260 20 μM, GKT137831 10 μM, Yoda1 5 μM, GsMTx4 5 μM. Individual brightfield frames were extracted from the time series to visualize wound healing dynamics in different cell lines.

Barrier assays with IBIDI culture inserts: BxPC3 cells (DUOX1/2 WT, DUOX2 KO, DUOX1/2 KO) were seeded into 2-well culture inserts in a μ-dish (81176, IBIDI) as per manufacturer's guidelines. Once 100% confluency was reached in both wells, culture inserts were carefully removed, and medium was added to the dish. Imaging was performed using a Nikon Ti microscope with environmental controls maintained at 37 °C; images taken every 30 min for 24 h. For full assessment of monolayer dynamics, culture inserts were removed from the μ-dish and attached in the wells of a 24-well cell culture plate. Wells were filled with media and left overnight to ensure no leakage was occurring. Monolayers were cultured in the inserts and the wound closure assay/speed calculations were carried out as above. DUOX2 WT cells were measured in 3 experiments, DUOX2 KO and DUOX1/2 KO cell lines were measured in one experiment.

HyPer7-MEM wound healing assay: BxPC3 DUOX1/2 WT HyPer7-MEM cells were seeded into 8-well μ-slides (80826, IBIDI) and grown to confluency. Monolayers were imaged on an inverted Nikon Ti microscope with Yokogawa spinning disk integrated by Andor Fusion at 37 °C and 5% $CO_2$. Cells were imaged for 30 min pre-wounding at 1 min intervals, with the first 5 min discarded to allow for probe equalization. Imaging was paused and the monolayers were scratched with a P10 pipette tip on the microscope stage to maintain x/y positions. The first 3 h of wound healing was imaged every 30 s using a 10X objective. 5 independent experiments were carried out with 2 wells imaged per experiment, each with 3 fields of view per well. Separately, HyPer7 fluorescence was recorded for 9 h post wounding, imaged every 3 min using a 20X objective, with 2 wells per experiment, each with 3 fields of view for 5 independent experiments. HyPer7 was excited at both 405 nm and 488 nm and detected at 499–560 nm. Time series were processed as described for HyPer7 quantification. To produce average HyPer7 ratio values for wound healing an ROI was placed over the fullest extent of cell coverage at t = 0 (time of wounding). Mean pixel intensity was obtained for the full time series within this ROI, in ICY using the ROI Intensity Evolution plugin. ROI measurements were then used to track the change in area ($\mu m^2$) covered by the monolayer during cell front retraction and subsequent extension within the field of view. The mean HyPer7 ratio was normalized to $\mu m^2$ coverage of cells for each time point. Individual frames were extracted from both time series to visualize the change in HyPer7 488/405 nm ratio presented in graphs.

Calcium wave scratch assay: BxPC3 DUOX1/2 WT cells were seeded into 8-well μ-slides (80826, IBIDI) and grown to confluency. Monolayers were imaged on an inverted Nikon Ti microscope with Yokogawa spinning disk integrated by Andor Fusion at 37 °C and 5% $CO_2$. 5 μM Fluo4-AM was added to the media for 45 min before imaging commenced. Initially, the confluent monolayer on mounted slides was imaged. Then, the monolayer was scratched with a p10 pipette tip while imaging continued to account for the rapid speed of calcium signaling, occurring in 0–30 s. This resulted in a brief loss of signal at the time of scratching but the signal returned to the correct focal plane after the pressure of the pipette tip was released. Scratched monolayers were then imaged for 2–5 min. The change in bound calcium by Fluo4-AM was quantified by placing an ROI of the fullest extent of cell coverage at t = 0 (time of wounding). Mean pixel intensity was obtained for the full time series within this ROI by using the ROI Intensity Evolution plugin in ICY. Fluo4-AM quantification is shown from a representative single experiment of a total 6 biological replicates, as displaying the mean was not feasible due to variation in scratching combined with the speed of the biological response. The same trend of increased Fluo4-AM intensity wave after scratching was

observed in each replicate. Fluo4-AM scratch assay images and movies are presented as a pseudo-color red heatmap to differentiate it from HyPer7-MEM heatmaps.

## Permeability

Monolayer permeability was measured by FITC dextran (TbD Labs) transwell assays. BxPC3 cells (DUOX1/2 WT and DUOX1/2 KO) were seeded into Millicell hanging cell culture inserts (3 μm diameter pores; PTSP24H48, Merck Millipore) inserted into 24-well culture plates. Once cells had reached full confluency, medium in both compartments was replaced with fresh medium. Medium placed in the upper compartment contained 60 μM 4 kDa FITC dextran. Plates were returned to the incubator for 2 h (37 °C, 5% $CO_2$). 100 μl of medium was taken from each of the lower wells and fluorescent intensities were measured in triplicate using a Synergy MX plate reader (Biotek). Concentration of FITC dextran was quantified by using a standard curve. After measurement of undamaged confluent monolayers, both upper and lower wells were washed with PBS, and incubated in fresh medium for 1 h at 37 °C, 5% $CO_2$. Monolayers were then scratched with a P10 pipette tip, and the above procedure was repeated (measurements taken at 2 h post wounding). Further measurements were taken at 26 h and 50 h post wounding. Cells were washed and fresh medium was applied after each measurement. Experiments were repeated 5 times, with 3 replicates per cell line. Non-scratched monolayers and empty wells were used as controls.

## F-actin quantification

F-actin was quantified using the G-actin/F-actin In Vivo Assay Kit (Cytoskeleton Inc. BK037) as per manufacturer's protocol. $1 \times 10^6$ cells were seeded into 6-well cell culture plates and allowed to attach overnight. Media was exchanged for warm (37 °C) 100 μl LAS01 buffer (Lysis and F-actin Stabilization Buffer). Cells were harvested by scraping, lysate was pipetted into 1.5 mL tubes. Samples were homogenized by vortex and repeated pipetting with a P20 tip. Lysates were incubated at 37 °C for 10 min and centrifuged for 5 min at 350 g at RT. Supernatant was ultracentrifuged at $100,000 \times g$ for 1 h at 37 °C. Supernatant containing total G-actin was transferred to fresh tubes, while the F-actin pellet was resuspended in actin depolymerization buffer and incubated for 1 h on ice. Both G-actin and F-actin samples were then processed for immunoblots that were probed with anti-Actin mouse monoclonal antibody (Cytoskeleton; 7A8.2.1). ImageJ was used to process the blot images and to calculate F-actin as a % of total actin in each sample.

## Quantitative PCR

BxPC3 DUOX1/2 WT cells were seeded into 6-well cell culture plates at a concentration of $3 \times 10^6$. 24 h after seeding, cells were washed, lysed in RLT buffer (79216, Qiagen) and detached by scraping. RNA extraction was carried out using the RNeasy Mini kit (74104, Qiagen). RNA concentration and purity was assessed by nanodrop ND-1000 spectrometer (ThermoFisher) The High-Capacity cDNA reverse transcription kit (10400745, Applied Biosystems) was used to produce cDNA at a concentration of 50 ng/μl. Real time PCR was performed in Micro-Amp™ Optical 384-Well Reaction Plate (10005724, Applied Biosystems) in a Quantstudio 7 Flex thermocycler (Applied Biosystems). The following TaqMan probes were used: DUOX1 Hs01047827-m1; DUOXA1 Hs00328806_m1; DUOX2 Hs00204187_m1; DUOXA2 Hs 01595311_g1; GAPDH Hs02786624_g1. Reverse transcriptase negative and $dH_2O$ samples were used as controls. Samples were run in triplicate for each gene on each plate run.

## Flow cytometry

NCI-H661 DUOXA2 cells were seeded at $3 \times 10^5$ cells per well in 12-well culture plates and transiently transfected with FuGene HD to express either DUOX2 WT, DUOX2 E843Q, or an empty vector, with 3 wells per construct. 48 h after transfection, cells were washed, trypsinized, and cell suspensions from the 3 wells were combined and counted. $1 \times 10^6$ cells in 200 μl FACS buffer (PBS with 2% FBS) were centrifuged (500 × g for 5 min) and resuspended in 100 μl blocking buffer (FACS buffer with 10% goat serum) for 15 min. Cells were centrifuged and resuspended in primary antibody solution (FACS buffer with anti-HA antibody 1:100 and eFluor780 live/dead stain 1:1000). Samples were incubated for 20 min at RT in darkness. Cells were washed 3 times, then resuspended in secondary antibody solution (FACS buffer with anti-mouse PE 1:100) and incubated for 20 min at RT in darkness. Cells were washed a further 2 times and finally resuspended in 400 μl FACS buffer. Flow cytometry analysis was carried out on an Accuri C6 Plus Flow Cytometer (BD Biosciences). Cells were first gated for singlets by forward and side scatter, then live/dead stain and PE to determine cells positive for HA tag without membrane permeabilization, indicating HA-DUOX2 plasma membrane localization. Non stained and secondary antibody only stained samples were used as controls for gating.

## Statistics and reproducibility

Statistical analyses were preformed using GraphPad Prism 8 (GraphPad Software, San Diego). Statistical test details are included in figure legends. Two group comparisons used unpaired $t$ tests with Welch's correction. For more than two groups analysis by Brown-Forsythe and Welch one way ANOVA and Dunnett's T3 post hoc test were used. $P$ values are shown for values higher than $P = 0.05$ in figure legends. No randomization or blinding was used. For HVA quantification, each point represents a single measurement of 3 technical replicates from 3 independent experiments, error bars ± standard deviation (SD). For kymograph analysis, each point represents a single lamellipodia from different cells across 3 independent experiments. For single cell migration velocity in BxPC3 and NCI-H661 cells, each point represents the average from 3 independent experiments with 20–25 cells tracked per experiment. Quantification of number of cells possessing at least 1 TNT between H661 cells, each point represents the average number of TNT possessing cells from 5 fields of view as a percentage of total number of cells within field of view. Experiments were repeated 4 times, each with 3 wells per cell line. In wound healing analysis, each point represents the average speed every 3 h, from 3 independent experiments, each with 3–6 replicates.

## Reporting summary

Further information on research design is available in the Nature Portfolio Reporting Summary linked to this article.

## Data availability

All data supporting the results of this study are available in the main text or in Supplementary Information. Source data are provided with this paper.

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

## Acknowledgements

We thank the UCD Conway Imaging Core facility, J. Simpson (UCD) for UnaG microscopy advice, I. Sorrentino (University Carlos III Madrid) for advice on HyPer7 imaging, and O. Blaque (UCD) for antibodies. Figures 1a,e,m, 2j, 6a were created with BioRender.com. This research was supported by Science Foundation Ireland (16/IA/4501, 22/FFP-A/10349, to U.G.K.) and a Government of Ireland Postgraduate Scholarship (GOIPG/2019/55 to M.O'M.).

## Author contributions

Design/conceptualization by U.G.K. and M.O'M. M.O'M. performed all experiments and data analysis. Cloning/CRISPR was performed by S.Z. Funding acquisition, project administration and supervision by U.G.K. M.O'M., and U.G.K. wrote the initial and final drafts and generated the figures.

## Competing interests

The authors declare no competing interests.
