## [Transparent Peer Review file · Nature Communications]

Spatiotemporal H₂O₂ flashes coordinate actin cytoskeletal remodeling and regulate cell migration and wound healing

Corresponding Author: Professor Ulla Knaus

Version 0:

Reviewer comments:

Reviewer #1

(Remarks to the Author)

This is a well-written manuscript that addresses the participation of the NADPH oxidase homolog DUOX2 on cell dynamics and migration. The authors used an elegant system using fluorescently tagged DUOX2 (with UnaG) and a newly generated membrane-targeted HyPer7-MEM probe in live-cell imaging approaches to show dynamic changes in DUOX2 localization with in association with increased H₂O₂ production, particularly in the context of tunnelling nanotube formation and cell migration dynamics. With respect to the latter, they associate DUOX2 with protruding lamellipodia but also at the rear retraction zone. They linked this with localized activation of the tyrosine kinase FER and phosphorylation of cortactin, as a key mechanism of actin polymerization, although the exact oxidative mechanism (e.g. protein cysteine target that controls FER activation) was not identified. Finally, the authors demonstrated that DUOX2 primarily participates in the initial retraction wave in wound healing assays. In general, even though a number of other studies have previously linked NOX enzymes to wound healing and cell migration in various cell types and organisms, the present studies importantly extend these findings, particularly by more definitively establishing localized DUOX2 presence and activation in the context of TNT formation or cell migration, thereby further highlighting the complex nature of localized NOX-dependent H₂O₂ signals impacting distinct features of e.g. cell migration. However, a few points deserve more clarification or discussion.

Specific comments:

1. The HyPer7-MEM approach showed that H₂O₂ production was associated with DUOX2 in formation of TNTs, but no data are actually shown to directly demonstrate that H₂O₂ production was responsible for TNT formation, as was suggested. Without such direct evidence, it cannot be ruled out that DUOX2 might act by an alternative mechanism, for example by local formation of NADP⁺ or NAADP⁺ (e.g. PMID: 34784249), which would deserve some consideration
2. One limitation of this work is that it is solely based on cancer cells and/or overexpression systems, and it is therefore questionable to what extent these findings are relevant for normal differentiated epithelial cells that primarily express other NOX homologs (DUOX1, NOX1, etc), instead of or in addition to DUOX2. Are effects of different NOX enzymes within a given cell type expected to be redundant or complementary? This aspect was not well discussed, and deserves more consideration.
3. The presented studies nicely illustrate the localized H₂O₂ bursts in association with e.g. cell migration, but there was no discussion with respect to the actual oxidative event(s) by which DUOX2 mediates e.g. migration or cell front retraction. Is Fer itself the oxidized target, or is its activation instead linking to oxidative inactivation of a phosphatase? Alternatively, could it be related to an upstream kinase such as Src, which has been demonstrated to be redox sensitive, and may also be involved in Fer activation? At a minimum, these issues deserve some more discussion.

Minor comments:

1. The authors nicely implicate PIEZO as a mediator of Ca²⁺ signaling, but did not actually demonstrate Ca²⁺ fluxes, which would have strengthened the studies.
2. The authors show that WT BxPC3 primarily express DUOX2 upon stimulation with IFN/LPS, but appear to have examined the roles of DUOX2 in lamellipodia formation and cell migration in the absence of IFN/LPS (at least this was not explicitly

stated). This begs the question whether these DUOX2-dependent processes might be enhanced upon induction of DUOX2 by IFN/LPS. Did the authors address this?

3. Fig. 3 only describes the formation of the BxPC3 cells lacking DUOX1/2, but does not seem essential as a regular manuscript figure, and could be moved to the supplementary figures.

Reviewer #2

(Remarks to the Author)

The study provides valuable insights into how "redox flashes" orchestrate cytoskeletal reorganization during processes such as cell migration, cell-cell communication, and wound healing. While the involvement of NADPH oxidases in cell motility has been previously documented, the authors present a dynamic and vesicle-based mechanism for the localization of the NADPH oxidase DUOX2 to sites of active actin remodeling. Upon activation, DUOX2 produces localized bursts of H₂O₂, which regulate actin polymerization, cell migration, and intercellular communication. The manuscript demonstrates that DUOX2-containing vesicles are transported to areas of active cell movement, such as lamellipodia, cell edges, and tunneling nanotubes, where H₂O₂ generation modulates key signaling pathways including tyrosine kinases like FER, which in turn activates downstream targets like cortactin. The discovery of a PI3EZO1-DUOX2-FER axis underlying the biphasic wound-healing response (retraction then closure) has wide implications for repair of damaged barrier epithelia. The potential for publication in Nature Communications is high but several major and minor points will need to be addressed:

Major points:

(1) The study highlights the spatiotemporal dynamics of H₂O₂ generation by DUOX2 at specific cellular sites like protrusions, TNTs, and lamellipodia. Even though DUOX2 is strongly implicated in H₂O₂ production at these sites, localized H₂O₂ bursts may also arise from other H₂O₂ producing enzymes. To clarify the contribution of DUOX2, the authors could investigate whether the HyPer7-MEM signal at plasma membrane protrusions is reduced in DUOX1/2 KO BxPC3 cells.

(2) The colocalization of DUOX2, FER, and pCortactin, along with the observed interaction between DUOX2 and FER, suggest a potential functional interaction between these two proteins at critical sites for cytoskeletal rearrangements. Inhibition of FER reduced single-cell migration velocity (Fig. 5d), decreased cell front speed and abolished the retraction wave during epithelial wound healing (Fig. 7b and c), which phenocopied DUOX1/2 KO cells, further supporting DUOX2 function in the same pathway. However, additional genetic data are needed to verify the downstream role of FER in DUOX2-mediated actin remodeling.

The authors show that inhibition of FER in DUOX1/2 WT BxPC3 cells led to a decrease in single-cell migration velocity, while the authors assessed tyrosine phosphorylation of FER in H661 cell lysates. Compared to H661 cells lacking DUOX2, H661 cells expressing activated DUOX2 displayed a moderate increase in FER tyrosine phosphorylation, though this was not statistically significant. The authors suggest that this lack of significance could be attributed to the fact that "whole cell lysates cannot adequately reflect this dynamic situation." However, given that H661 cells do not express DUOX1/2 or other NADPH oxidases, and do not produce H₂O₂ unless DUOX2 is lentivirally transduced (Fig. 1C), it is noteworthy that H661 cells lacking DUOX2 still exhibit relatively high basal levels of FER tyrosine phosphorylation (Fig. 5e). This raises the possibility that H661 epithelial cells may activate FER through H₂O₂-independent mechanisms. Therefore, it would be necessary to investigate whether DUOX1/2 KO BxPC3 cells show reduced tyrosine phosphorylation of FER compared to WT BxPC3 cells.

To confirm that DUOX2 activates FER, which in turn phosphorylates cortactin, the authors can also compare the local phosphorylation levels of cortactin at the leading edge between WT and DUOX1/2 KO BxPC3 cells, or between H661 cells without DUOX2 and with activated DUOX2.

Minor points:

(1) Given the substantial amount of data presented, it would be helpful to include a final schematic summarizing how DUOX2-mediated vesicle trafficking, localized bursts of H₂O₂, the mechanosensor PIEZO1, FER kinase activation, and the "retraction wave" converge. This would aid in integrating all of the findings and provide a clearer, more unified model of the signaling pathways involved.

(2) In figure legends of Fig. 1c, g, Fig. 2d, Fig. 3c, f, Fig. 4e-g, k, and Fig. 5f, the notation "error bars \pm SD" may be confusing. It would be clearer to state: "Data are presented as mean \pm SD"

(3) For Fig. 5d, i, j, are the error bars representing SD?

(4) For the data points with error bars in Fig. 6b, c, e, f, g and Fig. 7b, c, f, j, could the authors confirm whether the error bars represent SD? Additionally, are the error bars absent or at zero for some data points because SD = 0?

Version 1:

Reviewer comments:

Reviewer #1

(Remarks to the Author)

The authors have only partially responded to my original comments. For example, they did not fully address my initial comment regarding the critical role of H₂O₂ in e.g. TNT formation, and only provided indirect arguments. In the absence of

direct experimental evidence that functional outcomes associated with DUOX2 were indeed due to H₂O₂, the authors should acknowledge the possible involvement of additional alterations associated with DUOX2 activation, such as oxidation of NADPH or NAADPH, or local pH changes leading to activation of voltage-gated H⁺ channels. Second, while it is understandable that the authors used cell lines to address the fundamentals of localized DUOX2, their discussion of the biological significance was a bit underwhelming. In most epithelia, other NOXes have been implicated in wound healing responses rather than DUOX2. Also the potential significance for increased DUOX2 in pancreatic cancer was not discussed.

Reviewer #2

(Remarks to the Author)

I have reviewed the updated manuscript, and I am pleased to see that all of my concerns have been adequately addressed. The changes made improve the overall quality of the paper, and I believe it is now suitable for publication.

Version 2:

Reviewer comments:

Reviewer #1

(Remarks to the Author)

The authors have satisfactorily responded to my previous comments, and I have no further concerns.

REVIEWER COMMENTS

Reviewer #1 (Remarks to the Author):

This is a well-written manuscript that addresses the participation of the NADPH oxidase homolog DUOX2 on cell dynamics and migration. The authors used an elegant system using fluorescently tagged DUOX2 (with UnaG) and a newly generated membrane-targeted HyPer7-MEM probe in live-cell imaging approaches to show dynamic changes in DUOX2 localization in association with increased H₂O₂ production, particularly in the context of tunnelling nanotube formation and cell migration dynamics. With respect to the latter, they associate DUOX2 with protruding lamellipodia but also at the rear retraction zone. They linked this with localized activation of the tyrosine kinase FER and phosphorylation of cortactin, as a key mechanism of actin polymerization, although the exact oxidative mechanism (e.g. protein cysteine target that controls FER activation) was not identified. Finally, the authors demonstrated that DUOX2 primarily participates in the initial retraction wave in wound healing assays. In general, even though a number of other studies have previously linked NOX enzymes to wound healing and cell migration in various cell types and organisms, the present studies importantly extend these findings, particularly by more definitively establishing localized DUOX2 presence and activation in the context of TNT formation or cell migration, thereby further highlighting the complex nature of localized NOX-dependent H₂O₂ signals impacting distinct features of e.g. cell migration. However, a few points deserve more clarification or discussion.

We thank the reviewer for carefully reviewing our manuscript and the positive remarks.

Specific comments:

1. *The HyPer7-MEM approach showed that H₂O₂ production was associated with DUOX2 in formation of TNTs, but no data are actually shown to directly demonstrate that H₂O₂ production was responsible for TNT formation, as was suggested. Without such direct evidence, it cannot be ruled out that DUOX2 might act by an alternative mechanism, for*

example by local formation of NADP+ or NAADP+ (e.g. PMID: 34784249), which would deserve some consideration.

We show in Figure 2d that introducing a point mutation in the DUOX2 calcium binding EF hand, thereby preventing calcium binding and H₂O₂ production (Fig. 1c), decreased markedly TNT formation in cells. The above cited publication (2021) proposed that DUOX2 and DUOX1 bind not only NADPH but also NAADPH during TCR/CD3 stimulation of T cells to increase calcium microdomains, a process deemed independent of DUOX generated H₂O₂. It has not yet been clarified how these results harmonize with the long-standing notion that DUOX enzymes require a rapid increase of intracellular calcium for their initial activation, and if this scenario can be translated to other cell types when the cytosolic NADPH concentration is approx. 3 μ M while the NAADPH concentration is \leq 40nM. T cell DUOX-mediated NAADP generation was shown in vitro using cell membranes or by incubation of T cells with the NAADP antagonist BZ194 used at 500 μ M in 1% DMSO overnight. The prolonged incubation time with these high concentrations of BZ194 and DMSO is not tolerated by most cell types including the epithelial cells used in our study. To further address the reviewer's question about the importance of H₂O₂ for TNT formation, GKT137831, an NADPH oxidase inhibitor, was used as we demonstrated in 2014 that this chemical eliminated measurable DUOX1/2 generated H₂O₂ at a concentration of 10 μ M without showing any toxicity (Strengert, Knaus, 2014, Fig 5C, <https://doi.org/10.1089/ars.2013.5353>). Treatment of H661 cell lines (DUOX2 neg, DUOX2 WT, and DUOX2 E843Q) with GKT sharply decreased DUOX2 WT induced TNT formation, demonstrating that not only calcium binding but also H₂O₂ is required for DUOX2-facilitated TNT formation (see new Figure 2d). In all experiments except for the initial BxPC3 cell line characterisation (Fig. 3) and Fig. 5 e,f no exogenous stimulus was provided to increase the calcium concentration or to induce DUOX2/DUOXA2 upregulation.

2. One limitation of this work is that it is solely based on cancer cells and/or overexpression systems, and it is therefore questionable to what extent these findings are relevant for normal differentiated epithelial cells that primarily express other NOX homologs (DUOX1, NOX1, etc), instead of or in addition to DUOX2. Are effects of different NOX enzymes within a

given cell type expected to be redundant or complementary? This aspect was not well discussed and deserves more consideration.

We used H661 lung epithelial cells for overexpression studies as we showed that treatment with the demethylating agent 5-azacytidine leads to epigenetic reprogramming and expression of silenced DUOX1/2 (Luxen, Knaus, 2008, doi: 10.1158/0008-5472.CAN-07-5782), suggesting a relevant lung epithelial cell background. Pancreatic BxPC3 cells were used as they express endogenously catalytically active DUOX2/DUOXA2 that can be upregulated by cytokines and PAMPs similarly to primary cells (and is amendable to CRISPR knock out). While these two cell types are not primary cells, they reflect well endogenous DUOX2 expression in lung and pancreatic epithelium. Most, if not all, cell types express two or more NADPH oxidases (independently and/or dependent on stimulation). This fact is well known in the NOX field, yet signalling specificity of co-expressed NADPH oxidases has not been tackled in any comprehensive manner. We have shown distinct localization of DUOX1 vs DUOX2 in transfected cells and ALI differentiated primary lung epithelium (Luxen, Knaus 2009, doi: [10.1242/jcs.044123](https://doi.org/10.1242/jcs.044123)). This current study clearly demonstrates vesicle-localized DUOX2 in different cell types. Signalling specificity is, in our opinion, intimately linked to localization, but epitope tagging of NADPH oxidases is fraught with problems, usually sharply decreasing or abolishing their activity, or leading to mislocalization. As long as dynamic changes of the localization of several NADPH oxidases cannot be visualized in concert with their activation and ROS generation (superoxide, H₂O₂), signal specificity cannot be addressed. Initial generation of H₂O₂ by DUOX and NOX4 versus superoxide by NOX1-3,5 will, in our opinion, alter cellular redox signalling, while for example in the intestine NOX1 generated superoxide seems destined to produce ONOO⁻ in the presence of diffusible NO (Ward, Knaus 2023; Drieu La Rochelle, Knaus 2024). While compensation may occur when required (which is not per se redundancy), we hypothesize that NOX/DUOX signal specificity is dependent on localization, stimulus and cell type, similar as shown for Rho GTPases. This concept is now mentioned in the discussion.

3. The presented studies nicely illustrate the localized H₂O₂ bursts in association with e.g. cell migration, but there was no discussion with respect to the actual oxidative event(s) by which DUOX2 mediates e.g. migration or cell front retraction. Is Fer itself the oxidized target,

or is its activation instead linking to oxidative inactivation of a phosphatase? Alternatively, could it be related to an upstream kinase such as Src, which has been demonstrated to be redox sensitive, and may also be involved in Fer activation? At a minimum, these issues deserve some more discussion.

It has been established that the tyrosine kinase FER is activated by H₂O₂, leading to cortactin phosphorylation and cell migration. How H₂O₂ activates FER is still unknown although several approaches were taken to address this question (pers. communication P. Greer). In addition to linking DUOX2 generated H₂O₂ to FER and downstream cytoskeletal changes by using a specific FER inhibitor, and showing extensive co-localization of DUOX2 and FER, we now 1) co-localized DUOX2, FER and H₂O₂ (HyPer7 signal) in the same vesicles (new Fig.5a), and 2) knocked down FER with siRNA (new Supplementary Fig.5.d,e), followed by analysis of single cell migration (new Supplementary Fig.5f). Knockdown of FER significantly decreased single cell migration in BxPC3 DUOX2 WT cells. These experiments strongly support the DUOX2-FER connection in cell migration. Elucidating the molecular mechanism of FER activation by H₂O₂ flashes generated in DUOX2 containing vesicles is complex and awaits further study in the future.

Minor comments:

1. The authors nicely implicate PIEZO as a mediator of Ca²⁺ signaling, but did not actually demonstrate Ca²⁺ fluxes, which would have strengthened the studies.

We have now used the cell permeant calcium indicator Fluo-4 AM in wounding assays (new Supplementary Video 9, quantified in new Supplementary Fig.6f with example still images included in new Supplementary Fig.6g). We also show single cell calcium flux during membrane retraction and extension in the video (also new Supplementary Fig. 6h). As shown previously for PIEZO, mechanical wounding opened calcium channels and a wave of calcium was propagated from the cell front throughout the cell layer in a matter of seconds. Quantification of the Fluo-4 signal intensity is provided.

2. The authors show that WT BxPC3 primarily express DUOX2 upon stimulation with IFN/LPS,

but appear to have examined the roles of DUOX2 in lamellipodia formation and cell migration in the absence of IFN/LPS (at least this was not explicitly stated). This begs the question whether these DUOX2-dependent processes might be enhanced upon induction of DUOX2 by IFN/LPS. Did the authors address this?

As mentioned earlier, in all experiments except for initial BxPC3 cell line characterisation (Fig. 3) and Fig. 5 e,f no exogenous stimulus was provided to increase the calcium concentration or DUOX2/DUOXA2 expression. Stimulation of cells with IFN γ /LPS for 24h will alter multiple cell responses, i.e. both stimuli upregulate and modify many signalling pathways as upregulating DUOX2/DUOXA2 transcription is only one of the many facets of LPS/IFN γ signalling. Therefore, we purposely did not stimulate with IFN γ /LPS or PMA/thapsigargin/ionomycin in any of the actin cytoskeleton-driven processes. Only for the pTyr FER analysis stimulation with thapsigargin was used (as constitutive PKC activation can drive p-Tyr Cortactin). All experiments shown reflect DUOX2 activation and H₂O₂ production induced by cytoskeletal rearrangements and/or mechanical stimulation without any exogenous stimulus. Further, new Supplementary Fig.6h indicates dynamic intracellular calcium transients in vesicles during random migration without exogenous stimulation.

3. Fig. 3 only describes the formation of the BxPC3 cells lacking DUOX1/2, but does not seem essential as a regular manuscript figure, and could be moved to the supplementary figures.

If the reviewer agrees, we would like to keep the characterization of BxPC3 cells and their different DUOX CRISPR knockouts in the main Figure 3. This in-depth characterization is very important for the further conclusions of the manuscript, and as anybody will know working with this cell type, quite difficult to accomplish.

Reviewer #2 (Remarks to the Author):

The study provides valuable insights into how “redox flashes” orchestrate cytoskeletal reorganization during processes such as cell migration, cell-cell communication, and wound healing. While the involvement of NADPH oxidases in cell motility has been previously documented, the authors present a dynamic and vesicle-based mechanism for the localization of the NADPH oxidase DUOX2 to sites of active actin remodeling. Upon activation, DUOX2 produces localized bursts of H₂O₂, which regulate actin polymerization, cell migration, and intercellular communication. The manuscript demonstrates that DUOX2-containing vesicles are transported to areas of active cell movement, such as lamellipodia, cell edges, and tunneling nanotubes, where H₂O₂ generation modulates key signaling pathways including tyrosine kinases like FER, which in turn activates downstream targets like cortactin. The discovery of a PIEZO1-DUOX2-FER axis underlying the biphasic wound-healing response (retraction then closure) has wide implications for repair of damaged barrier epithelia. The potential for publication in Nature Communications is high but several major and minor points will need to be addressed:

We thank the reviewer for carefully reviewing our manuscript and the positive remarks.

Major points:

(1) The study highlights the spatiotemporal dynamics of H₂O₂ generation by DUOX2 at specific cellular sites like protrusions, TNTs, and lamellipodia. Even though DUOX2 is strongly implicated in H₂O₂ production at these sites, localized H₂O₂ bursts may also arise from other H₂O₂ producing enzymes. To clarify the contribution of DUOX2, the authors could investigate whether the HyPer7-MEM signal at plasma membrane protrusions is reduced in DUOX1/2 KO BxPC3 cells.

Efficient and stable expression of HyPer7-MEM in BxPC3 DUOX1/2KO cells by lentiviral transduction could not be accomplished (we do not know why as BxPC3 WT cells posed no problem; transient transfection of BxPC3 cells with HyPer7-MEM was also not sufficiently successful). Thus, we recorded HyPer7-MEM intensities at protrusions in H661 cells (+/- DUOX2). H661 cells expressing HyPer7-MEM were recorded on a spinning disk confocal

microscope at 20X magnification for 30 min with the first 10 min discarded for probe equalization. Time series were processed as per methods to calculate the HyPer7 488/405nm ratio. An ROI was placed over the fullest extent of a membrane protrusion of individual cells to generate the average pixel intensity of the HyPer7 ratio over time. The HyPer7 ratio signal, and thus H₂O₂, increased significantly as the membrane extended in H661 DUOX2 WT cells (new Supplementary Fig.2e; blue circles). The H₂O₂ flash can be seen for each extension at the four signal peaks, denoted by green arrows. The HyPer7 signal in H661 control cells (-DUOX2) was decreased, and membrane extensions did not generate a HyPer7 signal peak (new Supplementary Fig.2e; red triangles). This supports the substantial contribution of DUOX2 but no other H₂O₂ generating enzymes to membrane extension in our cell model.

(2)The colocalization of DUOX2, FER, and pCortactin, along with the observed interaction between DUOX2 and FER, suggest a potential functional interaction between these two proteins at critical sites for cytoskeletal rearrangements. Inhibition of FER reduced single-cell migration velocity (Fig. 5d), decreased cell front speed and abolished the retraction wave during epithelial wound healing (Fig. 7b and c), which phenocopied DUOX1/2 KO cells, further supporting DUOX2 function in the same pathway. However, additional genetic data are needed to verify the downstream role of FER in DUOX2-mediated actin remodeling.

In addition to linking DUOX2 generated H₂O₂ to FER and downstream cytoskeletal changes by using a specific FER inhibitor and showing extensive co-localization of DUOX2 and FER, we now 1) co-localized DUOX2, FER and H₂O₂ (HyPer7 signal) in the same vesicles (new Fig.5a), and 2) knocked down FER with siRNA (new Supplementary Fig.5.d, e), followed by analysis of single cell migration (new Supplementary Fig.5f). Knockdown of FER significantly decreased single cell migration in BxPC3 DUOX2 WT cells. We used a FER siRNA smart pool validated in several cell lines (Stanicka 2018, doi.org/10.1038/s41388-017-0113-z). These experiments strongly support the DUOX2-FER connection in cell migration.

The authors show that inhibition of FER in DUOX1/2 WT BxPC3 cells led to a decrease in single-cell migration velocity, while the authors assessed tyrosine phosphorylation of FER in H661 cell lysates. Compared to H661 cells lacking DUOX2, H661 cells expressing activated

DUOX2 displayed a moderate increase in FER tyrosine phosphorylation, though this was not statistically significant. The authors suggest that this lack of significance could be attributed to the fact that "whole cell lysates cannot adequately reflect this dynamic situation." However, given that H661 cells do not express DUOX1/2 or other NADPH oxidases, and do not produce H₂O₂ unless DUOX2 is lentivirally transduced (Fig. 1C), it is noteworthy that H661 cells lacking DUOX2 still exhibit relatively high basal levels of FER tyrosine phosphorylation (Fig. 5e). This raises the possibility that H661 epithelial cells may activate FER through H₂O₂-independent mechanisms. Therefore, it would be necessary to investigate whether DUOX1/2 KO BxPC3 cells show reduced tyrosine phosphorylation of FER compared to WT BxPC3 cells.

As suggested, we performed immunoprecipitation of FER in BxPC3 cells followed by pTyr detection and quantification (new Supplementary Fig.5g), indicating increased pTyr FER phosphorylation in stimulated conditions. Basal levels of pTyr FER in whole cell lysates were low, and although there was an increase in pTyr FER in stimulated BxPC3 DUOX1/2 WT cells it was not yet statistically significant, a similar result as observed in H661 cell lines. Yet, considering the typical loss of pTyr phosphorylation during lysate preparation and immunoprecipitation, the distinct localization of DUOX2, FER, and H₂O₂ in BxPC3 cell vesicles (new Fig.5a), and the brief duration of H₂O₂ flashes observed in videos, our hypothesis that "whole cell lysates cannot adequately reflect this dynamic situation" seems to be well founded.

To confirm that DUOX2 activates FER, which in turn phosphorylates cortactin, the authors can also compare the local phosphorylation levels of cortactin at the leading edge between WT and DUOX1/2 KO BxPC3 cells, or between H661 cells without DUOX2 and with activated DUOX2.

We quantified the local phospho-cortactin levels in H661 cells using confocal microscopy. 20X images demonstrated a stark difference in pCortactin staining at membrane protrusions in DUOX2 positive versus DUOX2 negative cells (new Supplementary Fig.5k). Individual cells were imaged at 60X and pixel intensity histograms were generated for ROIs placed around membrane protrusions, demonstrated in new Supplementary Fig.5l. Quantification of these histograms showed a significantly higher number of high intensity pixels in the membrane

protrusions of DUOX2 expressing cells (new Supplementary Fig.5m). Pixel intensity correlated to an increase in fluorescent antibody bound to the primary target, indicating an increase of phosphorylated Cortactin in membrane protrusions when active DUOX2 is present. Although the levels of total Cortactin remained similar between both cell lines, active generation of H₂O₂ by DUOX2 increased the amount of phosphorylated Cortactin (new Supplementary Fig5l).

A similar scenario is present in BxPC3 cells, see below. Due to space limitations, we could not incorporate these images into the Supplementary Fig. 5.

Phospho-Cortactin (red) staining in BxPC3 cell lines. Imaged at 20X magnification, scale bar 100 μ m.

Minor points:

(1) *Given the substantial amount of data presented, it would be helpful to include a final schematic summarizing how DUOX2-mediated vesicle trafficking, localized bursts of H₂O₂, the mechanosensor PIEZO1, FER kinase activation, and the “retraction wave” converge. This would aid in integrating all of the findings and provide a clearer, more unified model of the signaling pathways involved.*

We thank the reviewer for this suggestion as it will provide a nice overview of DUOX2/H₂O₂ activities in wound healing. The scheme is included as new Fig. 7k.

(2) *In figure legends of Fig. 1c, g, Fig. 2d, Fig. 3c, f, Fig. 4e-g, k, and Fig. 5f, the notation “error bars \pm SD” may be confusing. It would be clearer to state: “Data are presented as mean \pm SD”*

(3) *For Fig. 5d, i, j, are the error bars representing SD?*

As suggested by the reviewer, figure legends have been changed to "Data are presented as mean \pm SD".

Fig. 5d, i, j error bars represent SD. The figure legend has been updated to reflect this.

(4) For the data points with error bars in Fig. 6b, c, e, f, g and Fig. 7b, c, f, j, could the authors confirm whether the error bars represent SD? Additionally, are the error bars absent or at zero for some data points because SD = 0?

Error bars for Fig.7j are \pm SEM, the figure legend was changed to reflect this.

For Fig. 6b, c, e, f, g and Fig.7b, c, f data are presented as mean only. Presenting \pm SD was originally considered but due to the number of points and different conditions/lines on each graph, including the error bars would make the graphs difficult to read and obscured the message. An example of wound front velocity during PIEZO1 stimulation is provided below to highlight this point.

Second response to Reviewer 1

The authors have only partially responded to my original comments. For example, they did not fully address my initial comment regarding the critical role of H₂O₂ in e.g. TNT formation, and only provided indirect arguments. In the absence of direct experimental evidence that functional outcomes associated with DUOX2 were indeed due to H₂O₂, the authors should acknowledge the possible involvement of additional alterations associated with DUOX2 activation, such as oxidation of NADPH or NAADPH, or local pH changes leading to activation of voltage-gated H⁺ channels.

We do not understand this reasoning and disagree with the reviewer. The reviewer asked in the first review:

The HyPer7-MEM approach showed that H₂O₂ production was associated with DUOX2 in formation of TNTs, but no data are actually shown to directly demonstrate that H₂O₂ production was responsible for TNT formation, as was suggested. Without such direct evidence, it cannot be ruled out that DUOX2 might act by an alternative mechanism, for example by local formation of NADP⁺ or NAADP⁺ (e.g. PMID: 34784249), which would deserve some consideration.

We show as much as conceivably possible direct experimental evidence that calcium-activated DUOX2 produces H₂O₂ in vesicles containing DUOX2, HYPER7-MEM as H₂O₂ sensor and FER, and that DUOX2-generated H₂O₂ is required for cytoskeletal dynamics, TNT formation, migration and wound healing. See below:

1) Single cell calcium flux during membrane retraction and extension (Video and Supplementary Fig. 6h). Further, after wounding a wave of calcium was propagated from the cell front throughout the cell layer in a matter of seconds (see Supplementary Figure 6f, g, Supplementary Video 9). Thus, increased calcium, a prerequisite for DUOX2 activation, is present.

2) Figure 2d: introducing a point mutation in the DUOX2 calcium binding EF hand, thereby preventing calcium binding and H₂O₂ production (Fig. 1c), decreased markedly TNT formation in cells. Thus, calcium binding to induce DUOX2 activity is required for TNT formation.

3. DUOX2 and H₂O₂ are co-localised at the tip of TNTs during formation, inside TNTs and at the tip of elongating TNTs. Thus, enzyme and enzymatic product are co-localized in space and time (Figure 2, Supplementary videos 4).

3. The antioxidant GKT137831 eliminates oxidant production by all NADPH oxidases including DUOX1 and DUOX2 as outlined with citation in our initial response (Comments: GKT137831 is not considered a specific oxidase inhibitor anymore, but without overt toxicity the compound seems not to affect the mitochondria; specific oxidase isoform inhibitors do not yet exist; we did not use non-permeable catalase as DUOX2 generated H₂O₂ is localized inside vesicles). Treatment of H661 cell lines (DUOX2 neg, DUOX2 WT, and DUOX2 E843Q) with GKT137831 sharply decreased TNT formation in DUOX2 WT H661 cells, with no effect on the other two cell lines as they do not form a lot of TNTs due to absence of DUOX2 or due to absence of calcium binding to DUOX2. Thus, H₂O₂ is required for DUOX2 activity-facilitated TNT formation (Figure 2d). This experiment was done in response to the reviewer's initial request and was discussed in our first response to the Reviewer.

Thus, there is a direct line from 1)calcium, to 2)calcium binding to the enzyme, to 3)co-localization of enzyme and enzymatic product, to 4) need of enzymatic product (H₂O₂) for function, 5) observed H₂O₂-mediated function. If any of these steps are removed, TNT formation is markedly decreased. In conclusion, H₂O₂ is absolutely required for cytoskeletal dynamics including TNTs and wound healing, in agreement with the decades-long concept that NADPH oxidases are professional ROS generating enzymes. Our results are also 100% in line with the prevailing scientific notion on calcium-DUOX2 activity-H₂O₂-functional output.

In our first response to the Reviewer we pointed out that the publication *PMID: 34784249* cited by the Reviewer reported that NAADPH binding to DUOX2 and DUOX1 after TCR/CD3

stimulation of T cells increased calcium microdomains, and that this process was deemed H₂O₂ independent. This is obviously a completely different process that has not been demonstrated in any other condition, and, albeit new scientific concepts will always emerge, has at this point no connection to our study or our results.

Neither in the literature (except for *PMID: 34784249*) nor in our experiments can we reconcile '*oxidation of NADPH or NAADPH*' as mechanism for putative, H₂O₂ independent TNT formation. The HYPER 7 sensor is not pH sensitive, but it is well known that protons are being generated by activation of NOX2 in the phagolysosome which will lead to '*local pH changes leading to activation of voltage-gated H⁺ channels*' **and** superoxide/H₂O₂/HOCl generation. If this scenario would be present in DUOX2-H₂O₂ vesicles, it would still not account for our experimental result that removing H₂O₂ significantly decreases TNT formation. However, this scenario itself has been refuted by a very detailed study by Gattas and colleagues <https://doi.org/10.1016/j.redox.2019.101346> that came to the conclusion that the Hv1 channel is not connected to DUOX activity as Hv1 blockers inhibit other non Hv1 related processes.

We have shown in any conceivable way that calcium-activated DUOX2 produces H₂O₂ that is essential for TNT formation (and other cytoskeletal dynamics), a result that aligns with DUOX2 engagement and produced H₂O₂ as mediators of the actin cytoskeleton. We cannot follow why the reviewer labels our experiments as '*indirect*' argument and asks us to speculate on purely hypothetical scenarios. We cannot comment on scenarios that are either not present or highly speculative, just because one publication put forward an H₂O₂ independent mechanism for increasing calcium microdomains in stimulated T cells. As the journal will publish the Reviewer's comments and our responses, the scientific community will be aware that T cells can use DUOX in a non-canonical way.

Second, while it is understandable that the authors used cell lines to address the fundamentals of localized DUOX2, their discussion of the biological significance was a bit underwhelming. In most epithelia, other NOXes have been implicated in wound healing responses rather than DUOX2.

We have expanded the discussion regarding the role of NOX/DUOX in wound healing as requested. While several NADPH oxidases have been linked to single cell migration in the last two decades, either by using Boyden chambers or horizontal migration setups towards chemotactic stimuli, not many oxidases have been analysed in epithelial sheet migration, the process required for wound healing. A literature search shows a clear preponderance of H₂O₂ and DUOX1/2 as mediators, with NOX4 (H₂O₂) featuring in some cell types but not in others. Even though superoxide-generating NOX1 accelerated wound healing when overexpressed in COS cells (see in amended discussion), the mediator for cytoskeletal changes will be very likely H₂O₂ (colocalization of NOX1 with a peroxidase or rapid dismutation). In 2009 we compared the migration of COS cells expressing NOX4/p22 (H₂O₂) with COS cells expressing NOX2, p22, p47, p67 (termed COSphox, +/- PMA, superoxide), both cell lines producing the same output in nmol H₂O₂/h/mg protein. Migration of COS NOX4/p22 cells was substantially increased over COSp22 cells, while COSphox cells producing superoxide (PMA stimulation) migrated substantially less than COSphox cells without PMA stimulation or COSp22 cells (von Loehneysen, K et al, <https://doi.org/10.1128/MCB.01393-09>). As mechanical wound healing is often linked to an increase in the intracellular calcium concentration, it makes sense that calcium-activated DUOX has been much more prominent in wound healing scenarios.

Also the potential significance for increased DUOX2 in pancreatic cancer was not discussed.

As we outlined in the manuscript and in the first response, we used as one of the two cell models pancreatic cancer cells as these cells expressed endogenously DUOX enzymes, upregulation of DUOX2 followed published principles, H₂O₂ generation was fully dependent on calcium and DUOX2, cells were amenable to CRISPR, and cells migrated as sheets. Thus, these cells showed a phenotype comparable to primary cells that cannot be manipulated. Others in the field have shown that DUOX2 is highly expressed in certain cancers including pancreatic cancer. We do not refer to pancreatic cancer as there is scant literature on wound healing in the pancreas in the context of mechanical wounding and NADPH oxidase function. We added a general sentence about other cell types including cancer cells and NADPH oxidases with references.